# BloomQA: Automated Benchmark Generation from Domain Guidelines Informed by Bloom's Taxonomy

## Abstract

Open-ended question answering (QA) evaluates a model's ability to perform contextualized reasoning beyond factual recall. This challenge is especially acute in practice-based domains, where knowledge is procedural and grounded in professional judgment, while most existing LLM benchmarks depend on pre-existing human exam datasets that are often unavailable in such settings. We introduce BloomQA, a framework for automated benchmark generation from expert-authored guidelines infomed by Bloom's Taxonomy. BloomQA converts expert practices into implicit violation-based scenarios and expands them into auto-graded multiple-choice questions (MCQs) and multi-turn dialogues across four cognitive levels, enabling deterministic, reproducible, and scalable evaluation. Applied to three applied domains—teaching, dietetics, and caregiving—we find some differences between model and human-like reasoning: LLMs sometimes perform relatively better on higher-order reasoning (Analyze) but fail more frequently on lower-level items (Remember). BloomQA produces large-scale, psychometrically informed benchmarks that surface these non-intuitive model behaviors and enable evaluation of contextualized reasoning in real-world settings.

## 1 Introduction

Building open-ended QA benchmarks in new domains remains a fundamental challenge. Unlike mathematics and other structured fields—where problems can be systematically generated and large curated datasets exist—most applied domains lack readily available question banks. Practices such as teaching strategies or dietary counseling are inherently open-ended: they require situated reasoning, contextual knowledge, and adaptation to individual profiles. This makes it far more difficult to design benchmarks that probe reasoning beyond simple recall. These kind of practice-based domains (also referred to as procedural or applied domains), the primary knowledge involves knowing how to perform tasks, routines, and context-specific procedures [1].

Recent efforts have largely advanced by *crawling human-authored MCQs* from existing exams and certifications. For example, the Pedagogy Benchmark compiles ~1,000 items from Chilean teacher exams (Lelièvre et al., 2025), MedQA draws on U.S. and Chinese medical licensing exams (Jin et al., 2021), and FoodSky evaluates dietetics by mining professional certification questions (Zhou et al., 2024). While effective in narrow contexts, this strategy depends entirely on where high-quality exam banks already exist. As a result, benchmarks remain limited in size, costly to adapt, and unavailable for many practice-based domains where such resources, or certification exams, do not exist. The challenge is especially acute in practice-based domains, where expertise is demonstrated through contextualized reasoning rather than static fact recall. For example, in *dietetics*, FoodSky and related benchmarks show that even when exam data exist, Subject Matter Expert–LLM alignment remains partial, highlighting the difficulty of capturing applied expertise (Zhou et al., 2024; Szymanski et al., 2025). Together, these examples illustrate that relying solely on exam banks cannot adequately probe reasoning in practice-based domains.

We introduce BloomQA, a framework for constructing domain-grounded MCQs without relying on exam banks. Our approach uniquely combines LLM-assisted extraction of actionable best prac-

---

[1] https://en.wikipedia.org/wiki/Procedural_knowledge

tices from domain guidelines with systematic violation-based scenario generation. Each practice is verified by experts, then scripted into a violation scenario and converted into an MCQ with plausible distractors. Items are expanded into Bloom-taxonomy variants[2] (Remember, Understand, Apply, Analyze) and filtered through automatic checks and psychometric screening. This process produces practice- grounded MCQs that capture contextual reasoning and cognitive depth, enabling evaluation of nuanced decision-making skills beyond factual knowledge.

To gauge and evaluate the quality of the generated MCQs, we draw insights from educational measurement- a good test is defined not by the number of items, but by whether items discriminate ability, capture reasoning depth, and can confidently assess students with different ability fairly. Recent work in AI benchmarking has begun to adopt these psychometric notions—showing, for example, that small curated sets can be highly diagnostic (Srivastava et al., 2024) and that difficulty and discrimination analyses can meaningfully evaluate benchmark quality (Zhuang et al., 2025; Nguyen et al., 2025). These lessons motivate a shift from static exam repurposing to principled benchmark design.

Our research addresses two key questions: (1) Can we systematically generate high-quality MCQs from domain guidelines without human-authored exam banks? (2) Do these generated items effectively discriminate between different levels of reasoning ability in LLMs? This work makes three key contributions: (1) **A novel framework** (BLOOMQA) that transforms domain guidelines into psychometrically-validated MCQs through systematic violation-based scenario generation, eliminating dependence on existing exam banks. (2) **Three comprehensive benchmark datasets**, Teach-QA (19,824 items), Diet-QA (18,756 items), CareGiving-QA (20,000 items), each with Bloom-enriched variants that probe multiple cognitive levels. (3) **Empirical validation** demonstrating that our generated items achieve discrimination indices, establishing a scalable framework for benchmark creation in practice-based domains.

## 2 RELATED WORK

**Education Principles in AI Evaluation.** Recent work further adapts psychometric insights for AI benchmarking. For example, *TinyBenchmarks* shows that small, carefully chosen sets of items can be diagnostic (Srivastava et al., 2024). Zhuang et al. advocate for integrating item difficulty and adaptive testing into AI evaluation (Zhuang et al., 2025), while Nguyen et al. apply question quality analysis using simulated students (Nguyen et al., 2025). Together, there are growing interests in evaluation methods that emphasize discrimination, depth, and bias checks rather than raw accuracy.

**Bloom's Taxonomy in LLM Benchmarks.** Several benchmarks now apply Bloom's Taxonomy to assess LLM's cognitive depth. Huber and Niklaus (Huber & Niklaus, 2025) show that existing evaluations mostly target lower levels (Remember, Understand) with little coverage of higher skills (Evaluate, Create). *SciEval* (Sun et al., 2023) adopts Bloom to design multi-level scientific QA tasks, while *MedBloomEval* (Qiu et al., 2024) measures LLM reasoning across Bloom levels in medicine. BloomAPR introduces a dynamic Bloom's Taxonomy benchmark revealing LLM-based program repair excels at memorization but struggles with higher reasoning Ma et al. (2025). These efforts highlight the value of cognitive taxonomies and the lack study in practice-based domains.

**Three Example Domains** *Teaching and Pedagogy Domains.* Pedagogical expertise is difficult to benchmark due to its practice-based and context-dependent nature. Exam-derived resources such as the *Pedagogy Benchmark* remain limited in scope (Lelièvre et al., 2025). Recent efforts include *SimInstruct*, which models expert–novice scaffolding (Chen et al., 2025), and *MathDial*, which captures mathematics tutoring dialogues (Amini et al., 2023),but has limited generalization beyond individual subjects. *Food and Dietitian Domains.* Limited work was done in benchmarking though there are some on integrating graph learning with LLM reasoning for personalized, interpretable recommendations (Zhang et al., 2025) and *NGQA* models nutrition as graph-based health reasoning (Zhang et al., 2024). *Rehabilitation and Caregiving Domains.* Caregiving workflows remain highly disease-specific, limiting cross-condition transferability. Parmanto et al. developed a caregiving LLM grounded in an Alzheimer's-focused knowledge base (Parmanto et al., 2024). Yao et

---

[2]Bloom's Taxonomy is a hierarchical framework developed in 1956 that classifies learning objectives from basic recall of facts to higher-order skills like analyzing (Bloom et al., 1956).

al. introduced *DischargeSim*, simulating multi-turn doctor–patient discharge dialogues with diverse psychosocial profiles (Yao et al., 2025).

# 3 DATASET CONSTRUCTION

Our benchmark construction framework begins with domain practices, followed by violation scenarios that instantiate both structured MCQs and open-ended dialogues. Dialogues provide training material for fine-tuned models, while MCQs (and their Bloom enrichments) form the formal benchmark test. Both fine-tuned and existing models then take this test. Details of each construction stage are explained below, while analysis of test outcomes is presented in the Results section. Human expert's participation were under IRB-approved study protocols.

## 3.1 PRACTICES EXTRACTION FROM GUIDELINES

We present EXTRACTPRACTICES, an algorithm that leverages a large language model to automatically extract, structure, and filter actionable practices from raw text, applied similarly across domains with different PDF inputs.

---

**Algorithm 1** ExtractPractices: Core processing steps for diet as example

---
**Require:** Actionable paragraphs $P$, an LLM $\mathcal{M}$
**Ensure:** Structured and filtered practices
1:
2: **Structured Information Extraction**
3: **for** each $p \in P$ **do**
4:     $\widehat{p} \leftarrow \mathcal{M}(\text{prompt}_{\text{structure}} \| p)$ where $\widehat{p}$ contains:
5:         $\widehat{p}$.goal: health objective (e.g., "reduce heart disease risk")
6:         $\widehat{p}$.context: application setting (e.g., "grocery shopping")
7:         $\widehat{p}$.action: specific behavior (e.g., "choose whole grains")
8:         $\widehat{p}$.timing: frequency/schedule (e.g., "daily", "at breakfast")
9:         $\widehat{p}$.person: target demographic (e.g., "adults with diabetes")
10: **end for**
11:
12: **Granular Splitting**
13: **for** each $\widehat{p} \in P$ **do**
14:     **if** HasMultiplePractices($\widehat{p}$) **then**
15:         $P_{\text{split}} \leftarrow$ SplitIntoPractices($\widehat{p}$)
16:         $P \leftarrow (P \setminus \{\widehat{p}\}) \cup P_{\text{split}}$ {Replace with splits}
17:     **end if**
18: **end for**
19:
20: **Quality Evaluation and Deduplication**
21: **for** each $\widehat{p} \in P$ **do**
22:     $c \leftarrow \mathcal{M}(\text{prompt}_{\text{clarity}} \| \widehat{p})$ where $c \in [1, 5]$
23:     $s \leftarrow \mathcal{M}(\text{prompt}_{\text{similarity}} \| \widehat{p} \| P)$ where $s \in [1, 5]$
24:     **if** $c < 4$ **or** $s > 2$ **then**
25:         $P \leftarrow P \setminus \{\widehat{p}\}$ {Remove low-quality or redundant}
26:     **end if**
27: **end for**
28: **return** $P$ {Filtered and deduplicated practices} =0

---

The algorithm begins by extracting raw text from the document $D$ using ExtractText($D$) and dividing it into processable units through SplitParagraphs($T$) to create paragraph-level chunks $\mathcal{P}$. Each paragraph $p$ from $\mathcal{P}$ is evaluated by the LLM $\mathcal{M}$ using prompt$_{\text{filter}}$ to determine if it contains actionable food practices. Only paragraphs marked as actionable (where is_actionable $\neq$ "SKIP") are retained in the practice set $P$.

**Phase 1 - Structured Information Extraction (Lines 1-7):** For each actionable paragraph in $P$, the algorithm transforms it into a structured format $\widehat{p}$ using the LLM with prompt$_{\text{structure}}$ (Line 2). Each practice is tagged with five semantic components: goal (health objective), context (application setting), action (specific behavior), timing (frequency/schedule), and person (target demographic) as shown in Lines 3-7. It is informed by Five Ws (Who, What, When, Where, Why)- a checklist used in journalism to ensure that the lead contains all the essential points of a story [3].

---

[3]https://en.wikipedia.org/wiki/Five_Ws

**Phase 2 - Granular Splitting (Lines 9-14):** Complex paragraphs containing multiple practices are identified using HasMultiplePractices($\widehat{p}$) (Line 10) and decomposed into atomic practices through SplitIntoPractices($\widehat{p}$) (Line 11). The original complex practice is replaced with its split components (Line 12).

**Phase 3 - Quality Evaluation and Deduplication (Lines 16-22):** Each practice undergoes dual evaluation: clarity scoring $c$ (Line 17) and similarity scoring $s$ (Line 18) using specialized prompts. Practices with insufficient clarity ($c < 4$) or high redundancy ($s > 2$) are removed from the set (Lines 19-20). Here, the threshold 4 indicates that at least four of the five 5W elements (who, what, when, where, why) must be present and non-empty in a practice, while the threshold 2 indicates that no more than two 5W elements may be shared between any pair of practices.

After the algorithm returns the filtered practices (Line 23), summaries are generated using GenerateSummary($\widehat{p}$) and practices are organized by health goals through GroupByGoal($P$), producing the final dataset $\widehat{P}$ with approximately 60 structured practices. The retention condition for each practice $\widehat{p}$ is formally defined as:

$$\text{keep}(\widehat{p}) = \begin{cases} \text{true} & \text{if clarity}(\widehat{p}) \geq 4 \wedge \text{similarity}(\widehat{p}) \leq 2 \\ \text{false} & \text{otherwise} \end{cases} \tag{1}$$

Using above, we curate 60 dietary, 42 teaching, 62 caregiving practices, each describing a concrete recommended behavior (e.g., Limit alcoholic beverages to 2 drinks or less per day for men and 1 drink or less per day for women." Begin each class by revisiting students' previous predictions and guiding them to reflect on their accuracy and reasoning, fostering critical thinking and self-evaluation."). The dietary set is drawn from the *Dietary Guidelines for Americans, 2020–2025*, the teaching set is drawn from Lang's *Small Teaching* Book, and the caregiving set is drawn from *HOPE: The Stroke Recovery Guide*. In all domains, the practices were verified by domain experts with extensive publications in education, dietetics, mobility and human factors, ensuring that they are mutually exclusive, independently violable, and thus yield valid outcomes when transformed into benchmark items. Five dietary practices highly specific to children, 10 caregiving practices were highly specific to patient's self-care with a focus on disability mental status and deemed high-risk for scenario generation were filtered out, leaving 55 dietitian and 52 caregiving that were carried forward.

## 3.2 PRACTICES TO BENCHMARKS

Beelow is a three step process that leverages an LLM to generate violation scenarios, Bloom-enriched MCQs, and scaffolded dialogues from curated practices. The framework is applied consistently across domains once practices are obtained.

### 3.2.1 PRACTICES TO VIOLATION SCENARIOS

For each practice $p \in \mathcal{P}$, we generated a concise violation scenario $s$ (80–120 words) illustrating how $p$ was not followed. To avoid leakage, scenarios never mentioned the practice name directly; instead, violations were conveyed implicitly through actions and consequences (e.g., grading delayed for weeks" rather than no timely feedback"). Each scenario was grounded in a synthetic profile (e.g., a first-year college instructor teaching a large introductory course, or a middle-aged client trying to manage sugar intake) and created by the LLM, specially GPT-4o-mini, then filtered through rule-based automatic quality control for field presence, word-count limits, hash-based duplicate detection, and keyword/regex filters for unrealistic, unobservable or answer-leaking content (more in Appendix A.1.2). Model choice: Domain experts reviewing the post validation outputs did not identify systematic quality differences between GPT-4o-mini and Claude Sonnet 4. Human evaluation results are in Appendix A.1.3. Six teaching practices never passed automatic validation, leaving 36 teaching practices retained.

### 3.2.2 SCENARIOS TO MCQS AND BLOOM VARIANTS

Each retained scenario served as the basis for multiple-choice questions (MCQs). For Diet we used four options, and for Teaching five, with one correct option and the rest as plausible distractors from

the same domain. To incorporate cognitive depth, each base MCQ was expanded into four Bloom-aligned variants—*Remember*, *Understand*, *Apply*, and *Analyze*. This expansion produced 20,000 MCQs per domain from the the generated 5,000 scenarios. [4] Bloom's Taxonomy, first developed in 1956, provides a hierarchical framework for structuring learning objectives. By aligning MCQs to Bloom levels, we ensure the benchmark moves beyond recall to progressively deeper reasoning skills. This design choice is consistent with educational measurement literature, where four or five options per item and Bloom's taxonomy are regarded as best practices for valid, discriminative assessments (Haladyna & Downing, 1993; Vegada et al., 2016). Table 1 summarizes the guiding questions and option revisions.

| Bloom Level | Guiding Q. (constant) | How choices are revised | Example option |
|---|---|---|---|
| *Remember* | Which practice is violated in this scenario? | Keep original practice descriptions unchanged. | *Use retrieval practice to strengthen long-term memory.* |
| *Understand* | Which practice best explains why this challenge occurred? | Rephrase each option as a cause/effect explanation. | *Because retrieval practice was missing, students quickly forgot key concepts.* |
| *Apply* | Which practice should be used next time to address the problem? | Reframe each option as a forward-looking action. | *Add a brief retrieval quiz at the start of class to improve recall.* |
| *Analyze* | Which practice best fits this scenario compared to the others? | Expand each option with pros and cons to compare relevance across practices. | *Retrieval practice improves retention (pro), but may take extra grading effort (con).* |

Table 1: Bloom's MCQ enrichment. Guiding questions remain constant across scenarios and domains; choices are revised with one rule per level and illustrated with examples.

### 3.2.3 SCENARIOS TO DIALOGUES

To complement MCQs with process-level evidence, each scenario $s$ was extended into a multi-turn dialogue $d = \big((u_1, r_1), \ldots, (u_T, r_T)\big)$ between an instructor or client and an expert. Dialogues contained 20–30 turns and followed four phases: (i) problem understanding, (ii) solution exploration, (iii) implementation planning, and (iv) reflection. User turns were conditioned on the scenario profile to increase realism, and phases were scaffolded by Bloom's levels—*Remember*, *Understand*, *Apply*, *Analyze*—so that reasoning unfolded step by step while keeping the violation implicit. Each phase was allotted a target range of turns and enriched with Bloom-style scaffolding prompts so that reasoning unfolded progressively rather than in a single "answer dump." Responses were constrained to be natural (2–4 sentences), supportive, and actionable. In total, we generated 5,000 dialogues per domain, producing corpora for fine-tuning and evaluating open-ended instructional interactions. Bloom-scaffolded dialogues are valuable because they train models to act as helpful experts, engaging end-users with knowledgeable, stepwise teaching. They also approximate the structure and tone of authentic teaching, making them more realistic than isolated QA items. This step was implemented using GPT-4O-mini. Similarly, for dialogue generation, two models were used to generated the 150 conversations with the same prompt, and three human experts rated them, finding no significant differences in approximately 95% of cases.

### 3.3 SYNTHETIC DATASET DESCRIPTION AND SCALE

Our framework generated ∼5,000 scenarios and ∼5,000 dialogues per domain, with safeguards in place to ensure systematic coverage, avoid duplication, and validate realism. MCQ derived from scenarios are automatically verifiable through labeled correct answers, while dialogues are scaffolded to span 20–30 turns with realistic word counts per exchange, ensuring both depth and natural flow. Profiles tailored to each domain (e.g., dietary goals and constraints for Diet; class size and teaching experience for Teaching, caregiver-patient relationship for Caregiving) provided contextual grounding, and staged phases guided conversations through problem exploration, solution development, and reflection. Although demonstrated in Diet, Teaching and Caregiving, the framework is designed to be generalizable to any domain with a practice guidelines. Table 2 summarizes the dataset scale and dialogue characteristics.

**Human Evaluation**: From 5,000 generated scenarios that passed machine validation in each domain, random 50 scenarios was independently reviewed by three experts. Experts evaluated each

---

[4]We selected four Bloom levels because decades of educational measurement research show that the Evaluate and Create levels require open-ended reasoning and generative performance that cannot be validly assessed using MCQ Krathwohl (2002); Brookhart (2010).

Table 2: Dataset Statistics

| Dataset | #Practices | #Scenarios | #MCQs | #Dialogues | Turns / Dialogue | Words / Dialogue |
|---|---|---|---|---|---|---|
| Teach-QA | 36 | 5,000 | 20,000 | 4,987 | 21–32 (avg. 25.2) | 481–872 (avg. 645) |
| Diet-QA | 55 | 5,000 | 20,000 | 4,993 | 18–32 (avg. 25.3) | 419–833 (avg. 592) |
| CareGiving-QA | 52 | 5,000 | 20,000 | 5,000 | 20–32 (avg. 25.0) | 397–796 (avg. 572) |

item on *Violation Alignment* (exactly one clear practice violation), *Hint-Free* phrasing (no explicit answer leakage), and *Realism* (contextual plausibility). Conversation turns were additionally rated on *Conversational Progression* (logical, scaffolded flow) and *Repair Quality* (actionable, practice-aligned guidance). Inter-rater agreement was high (94%, 141/150), and overall acceptability was 96%. The small number of failures were primarily attributable to subtle answer leakage (e.g., phrasing such as "aren't sufficient"). To further validate quality, using Teaching domain as a cases, we recorded real consultations at a university teaching center and piloted a human-in-the-loop setup in which the LLM simulated clients and human experts provided guidance. Eight expert in teaching consultation judged these dialogues to be realistic and diverse, supporting the validity of the generated dataset.

## 4 DATASET EVALUATION METHOD

How do we know if our benchmark is a good test—and how might we improve it? We frame evaluation through a quiz analogy: *models are students*, *MCQs are questions*, and *Bloom levels capture layers of knowledge and skill*. For evaluation, we sampled 2,400 MCQs (600 scenarios $\times$ 4 Bloom levels) as the quiz set, and models then "took the test," as in a standard benchmark.

A benchmark, like a quiz, must be more than a score sheet. A good test should reliably separate strong from weak students, probe different layers of understanding built on prior knowledge, and give all students comparable changes (Downing & Haladyna, 2006). In the same way, an ML benchmark is only useful if it distinguishes model abilities, reflects progression from basic recall to more complex applications, and avoids misleading measurements when questions are poorly designed. Guided by this, our analysis plan addresses three core questions: 1. Does the benchmark probe multiple levels of domain knowledge according to Bloom's taxonomy (not just recall)? 2. Do questions effectively discriminate between strong and weak models, and are they free of confusing or poorly framed content? 3. Is the test fair and balanced, or does it disadvantage some models? [5]

**Step 1: Probe Bloom levels.** *Goal: Test whether the benchmark captures more than one levels of knowledge and skill, not just recall.* We fit binomial GLMMs with random intercepts for practices:

$$\text{logit}(p_{m,b}) = \beta_0 + \beta_{\text{Model}(m)} + \beta_{\text{Bloom}(b)} + u_{\text{Practice}}, \quad u_{\text{Practice}} \sim \mathcal{N}(0, \sigma^2).$$

Here, $p_{m,b}$ is the probability of correctness for model $m$ on Bloom level $b$. Coefficients for Bloom levels reveal whether performance differs systematically across layers of Bloom's taxonomy, confirming whether the benchmark spans from basic recall to more advanced applications.

Further, we computed the **Bloom Hierarchical Progression Rate** (BHPR) to quantify how learner performance at one Bloom level conditionally predicts performance at higher levels. Using conditional success–success (SGS) and success–failure (SGF) transition matrices, we estimated ordered relationships across Remember, Understand, Apply, and Analyze. In human learning, SGS is typically larger than SGF because success at a lower cognitive level strongly predicts success at higher levels, whereas failure at a lower level makes subsequent higher-level success much less likely.

**Step 2: Check Question's Model and Bloom Discrimination.** *Goal: Ensure both individual questions and Bloom levels separate strong from weak models, and flag poorly functioning questions.* Difficulty shows how hard an item is, while discrimination shows how well it separates stronger from weaker models; an item can be hard but not diagnostic. We use best linear unbiased predictions (BLUPs) from the GLMM to estimate per-model correctness on each practice. Item discrimination

---

[5]Statistical assumptions. Assumptions include conditional independence of trials and normally distributed practice effects. For reliable estimates, we recommend at least 30 practices per domain, balanced Bloom coverage, and about 1,000 trials, or MCQ in our case, per model. For discrimination and ranking analyses to be stable, we recommend benchmarking at least 6 models per domain.

is measured as

$$\Delta_{\text{model}} = \max_m \hat{p}_{m,p} - \min_m \hat{p}_{m,p},$$

where $\hat{p}_{m,p}$ is the predicted probability of correctness for model $m$ on practice $p$. Larger values of $\Delta_{\text{model}}$, such as above 0.5, mean stronger separation: For example, a student scoring 50/100 versus one scoring 100/100 would typically be viewed as meaningfully different ($\Delta_{\text{model}} = 0.5$), whereas the difference between 90/100 and 95/100 is often considered minor ($\Delta_{\text{model}} = 0.05$).

To test whether Bloom levels themselves drive meaningful variation, we compute

$$\Delta_{\text{bloom}} = \max_b \hat{p}_b - \min_b \hat{p}_b,$$

where $\hat{p}_b$ is the average probability of correctness at Bloom level $b$. In educational testing with human students, $\Delta$ above 0.2 are often taken as very meaningful: For example, for the same knowledge point, the same student scoring 70/100 versus 90/100 would typically be viewed as meaningfully different ($\Delta_{\text{bloom}} = 0.2$).

According to large-scale human student grade analyses Rojstaczer & Healy, most students receive A–B grades (80–100/100), while scores below 70 are generally treated as failing; thus, a 50-point gap (e.g., 100 vs. 50) represents a very strong, human-interpretable performance difference. Similarly, classical education measurement theory (the core behind testing system such as SAT, GRE) Hambleton & Swaminathan (1991); Embretson & Reise (2000); Shrock & Coscarelli (2013); Jensen & Likert (2017) treat difficulty shifts of 0.1–0.3 logits (roughly a 3–8% change in probability of a correct response) as meaningful, and shifts of 0.5 logits (roughly 15–20%) as major. This mapping motivates our use of $\Delta_{\text{model}} \approx 0.5$ and $\Delta_{\text{bloom}} \approx 0.2$ as human-grounded thresholds. For LLMs, however, the distribution of both $\Delta$ may deviate from the roughly normal pattern observed in human performance. Thus, $\Delta$ values remain informative for interpreting model behavior but should not be regarded as directly comparable to human students, accordingly, we visualized all $\Delta$ values (0–100) for both measures as an ablation check in the Appendix A.2 (Figure 3).

**Step 3: Reliability and Validity Checks.** *Goal: Verify that the benchmark gives all models comparable chances to demonstrate ability.* The dataset was designed with balanced Bloom coverage (25% per level) and equal exposure for each model. To check fairness, we compared observed versus expected correct counts for every Model×Practice (or Model×Practice×Bloom) cell under the GLMM. Large deviations (flagged when standardized residuals exceeded ±3 after Benjamini–Hochberg FDR correction at $p = 0.05$) indicate cases where a model did unexpectedly well or poorly on a localized subset of questions—much like a student suddenly acing a very hard quiz or failing an easy one.

## 5 EVALUATION RESULTS

### 5.1 OVERVIEW OF BENCHMARKING RESULTS ACROSS DOMAIN

We fit binomial generalized linear mixed models (GLMMs) with random intercepts for best practices (see Section 4 for model specification), estimating models separately for each domain. Substantial practice-level variance was observed across domains ($\sigma \approx 0.78$–$1.58$), confirming meaningful heterogeneity in item difficulty and justifying the use of clustered random effects. Full coefficient tables (estimates, SEs, $z$, and $p$ values) are reported in Appendix A.2 Figure 6. Both model identity and Bloom level were significant predictors of correctness. As shown in Table 3, in Teaching, the strongest performers were DEEPSEEK-V3, QWEN-25-72B, and KIMI-K2 (all $p \ll .001$), while LLAMA-33-70B and MIXTRAL-8×7B performed substantially worse. In Diet, KIMI-K2 clearly outperformed other models ($p \ll .001$), while LLAMA-33-70B and MIXTRAL-8×7B again lagged, and most remaining models clustered in the mid-range.

Bloom-level effects diverged across domains. In Teaching, Apply, Remember, and Understand were all significantly harder than Analyze (all $p < .001$). In Diet, Apply and Remember were significantly easier than Analyze (both $p < .001$), while Understand did not reliably differ. These results indicate that Bloom levels captured structured variation in difficulty, but with domain-specific directionality. Notably, whereas Bloom's taxonomy assumes increasing cognitive complexity at higher levels for human learners, LLMs showed uneven alignment with this hierarchy. To test domain effects directly, we fit a pooled GLMM with fixed effects for Domain, Model, and Bloom level

and random intercepts for Domain:Practice_ID. Adding Model×Domain interactions significantly improved fit over the main-effects model ($\Delta\chi^2(14) = 1609$, $p < 2 \times 10^{-16}$), confirming that models' relative strengths differ by domain. In the main-effects model, Diet did not differ significantly from Teaching, whereas Caregiving showed a substantially higher baseline log-odds of correctness ($\beta = 1.03$, $p < .001$), consistent with easier items in that domain. We further quantified BHPR using conditional transition matrices SGS SGF probabilities across all pairs of Bloom levels. Conditional on success at a lower level, success rates at higher levels remained high (SGS $\approx 0.87-1.00$), whereas conditional on failure, success at other levels remained low (SGF $\approx 0.00-0.34$). We further decomposed these transitions by direction (lower→higher vs. higher→lower) under both success- and failure-conditioned settings. Under SGS, higher→lower transitions were more likely than lower→higher, while under SGF the reverse pattern held (lower→higher > higher→lower); visualizations are provided in Appendix A.2(Figure 1 and 2). Together, these patterns suggest that LLM performance is directional consistent across Bloom levels as human learners, but does not follow the step-wise hierarchy.

| Model | Remember | | | Understand | | | Apply | | | Analyze | | | Avg. across Bloom | | |
|---|---|---|---|---|---|---|---|---|---|---|---|---|---|---|---|
| | Teach | Diet | CareG | Teach | Diet | CareG | Teach | Diet | CareG | Teach | Diet | CareG | Teach | Diet | CareG |
| DeepSeek-V3 | 0.699 | 0.646 | 0.604 | **0.730** | 0.586 | 0.723 | 0.595 | 0.610 | 0.727 | 0.904 | 0.585 | 0.825 | **0.732** | 0.607 | 0.720 |
| GPT-4O | 0.712 | 0.657 | 0.706 | 0.608 | 0.600 | 0.734 | 0.482 | 0.620 | 0.725 | 0.879 | 0.606 | **0.840** | 0.670 | 0.621 | 0.751 |
| GPT-4O-Mini | 0.717 | 0.700 | **0.721** | 0.662 | 0.561 | **0.771** | 0.618 | 0.653 | **0.770** | 0.850 | 0.615 | 0.828 | 0.712 | 0.632 | **0.772** |
| Kimi-K2 | 0.692 | 0.691 | 0.675 | 0.659 | **0.612** | 0.719 | **0.620** | **0.677** | 0.704 | 0.897 | **0.695** | 0.835 | 0.717 | **0.669** | 0.733 |
| Llama-33-70B | 0.512 | 0.629 | 0.708 | 0.299 | 0.523 | 0.735 | 0.215 | 0.595 | 0.704 | 0.427 | 0.547 | 0.833 | 0.363 | 0.574 | 0.745 |
| Mixtral-8×7B | 0.206 | 0.648 | 0.632 | 0.206 | 0.492 | 0.700 | 0.199 | 0.571 | 0.690 | 0.213 | 0.488 | 0.710 | 0.206 | 0.550 | 0.682 |
| Qwen-25-72B | **0.756** | 0.657 | 0.705 | 0.689 | 0.558 | 0.754 | 0.577 | 0.583 | 0.712 | **0.905** | 0.545 | 0.811 | **0.732** | 0.586 | 0.746 |
| Qwen-3-80B-Instruct | 0.669 | **0.729** | 0.644 | 0.630 | 0.574 | 0.709 | 0.575 | 0.649 | 0.697 | 0.852 | 0.598 | 0.800 | 0.682 | 0.638 | 0.712 |

Table 3: Accuracy by model and Bloom level, with Teaching vs. Diet vs. Caregiving (CareG) domains shown side-by-side. For Teaching and Caregiving, each MCQ had 5 options, so random guessing corresponds to 20% accuracy. For Diet, each MCQ had 4 options, so random guessing corresponds to 25% accuracy. Bold numbers highlight the highest-performing model in each column. Note: Numbers are not directly comparable across domains.

## 5.2 PRACTICES: DIFFICULTY AND DISCRIMINATION

Using baseline GLMM estimates (see Section 4), Teaching showed no below-chance practices, whereas Diet showed five flagged items and Cargiving has one (Table 7). Removing these items shifted the GLMM-estimated accuracies of benchmarked LLMs by at most $0.062$, yet the relative rankings of all models remained unchanged, confirming that overall conclusions are robust and not driven by a handful of pathological practices.

Most practices were discriminative. In Teaching, the median adjusted spread across models was $0.714$, with 35 of 36 practices (97.2%) showing strong separation. In Diet, the median spread was $0.509$, with 33 of 55 practices (60.0%) discriminative, similar to Caregiving. Our measure, $\Delta_{\text{model}}$, captures the difference between the weakest and strongest models on the same practice (definition in Section 4). For interpretation, we treat $\Delta_{\text{model}} \approx 0.20$ as the practical threshold for "meaningful separation," and values above $0.50$ as unusually strong. Similarly, Bloom separation was quantified as $\Delta_{\text{bloom}}$ (see Section 4). For Bloom, we treat $\Delta_{\text{bloom}} \approx 0.30$ as the threshold for meaningful cognitive differentiation, while values below $0.10$ suggest Bloom labels exert little influence. Using these measures, 26 of 36 Teaching practices (72.2%) and 20 of 55 Diet practices (36.4%)(similar to Caregiving) exhibited big Bloom effects or large spreads (Table 7). Full distributions of $\Delta_{\text{model}}$ and $\Delta_{\text{bloom}}$ are provided in Appendix 3. Manual qualitative inspection can be seen in Appendix A.2.

## 5.3 RELIABILITY AND VALIDITY CHECKS

We intentionally designed the dataset for balanced coverage along two axes: (i) depth exposure via Bloom mix and per-practice balance, and (ii) student exposure via trials per model. In Teaching and Diet, the Bloom mix was even (25% each for *Analyze*, *Apply*, *Remember*, and *Understand*), and every model had identical exposure (2,400 trials). To assess validity, we compared observed vs. expected correct counts for each *Model×Practice* (and optionally *Model×Practice×Bloom*) cell under the GLMM (see Section 4). At the practice level, Teaching had 7 better-than-expected and 31 worse-than-expected cells (38 of 288, 13.2%), Diet had 8 better and 5 worse (13 of 440, 3.0%), Caregiving

had 14 better and 6 worse (20 of 416, 4.8%). We visualized model fit by comparing observed versus expected counts across all Model×Pratice_ID cells (Figure 4). At the finer Bloom-expanded level, 52 of 1,152 Teaching cells (4.5%), 45 of 1,760 Diet cells (2.5%) and 41 of 1,664 Teaching cells (2.5%) were flagged. Mixtral-8X7B showed the largest performance swings, suggesting it behaves like a non-typical "student" whose ability may need to be assessed through a separate lens.

## 5.4 USE OF DIALOGUE DATA FOR MODEL FINETUNING

Table 4 reports model accuracies taking the same MCQ questions across Bloom's taxonomy levels in the Teaching, Diet, and Caregiving domains, comparing pre-trained models (top) with their fine-tuned counterparts (bottom). Fine-tuning was performed using our dialogue data, which was designed to strengthen domain knowledge through reasoning rather than by directly supplying answers. This approach provides a pathway for improving both domain grounding and the usefulness of the dataset itself. The results show consistent gains from fine-tuning, with the largest improvements in Teaching, especially at higher Bloom dimensions such as Apply and Analyze, where models like FT-Qwen2.5-14B achieve substantial increases. Gains are smaller but still present in Diet and Caregiving domain.

| Model | Remember | | | Understand | | | Apply | | | Analyze | | | Avg. across Bloom | | |
|---|---|---|---|---|---|---|---|---|---|---|---|---|---|---|---|
| | Teaching | Diet | Caregiving | Teaching | Diet | Caregiving | Teaching | Diet | Caregiving | Teaching | Diet | Caregiving | Teaching | Diet | Caregiving |
| Llama-3.1-8B | 0.165 | 0.317 | 0.280 | 0.065 | 0.045 | 0.318 | 0.128 | 0.183 | 0.273 | 0.290 | 0.092 | 0.303 | 0.162 | 0.159 | 0.294 |
| Qwen2.5-7B | 0.672 | 0.638 | 0.597 | **0.690** | 0.473 | 0.637 | 0.463 | 0.563 | 0.615 | 0.845 | 0.488 | 0.690 | 0.668 | 0.541 | 0.635 |
| Qwen2.5-14B | 0.603 | 0.638 | 0.592 | 0.590 | 0.507 | 0.602 | 0.445 | 0.565 | 0.625 | 0.840 | 0.532 | 0.692 | 0.620 | 0.561 | 0.628 |
| FT-Llama-3.1-8B | 0.194 | 0.333 | 0.329 | 0.087 | 0.062 | 0.426 | 0.156 | 0.193 | 0.333 | 0.303 | 0.114 | 0.317 | 0.185 | 0.176 | 0.351 |
| FT-Qwen2.5-7B | 0.662 | **0.660** | 0.598 | 0.688 | 0.497 | 0.643 | 0.487 | **0.578** | 0.627 | 0.845 | 0.513 | 0.695 | 0.671 | 0.562 | 0.641 |
| FT-Qwen2.5-14B | **0.703** | 0.648 | **0.628** | 0.673 | **0.517** | **0.683** | **0.507** | 0.570 | **0.655** | **0.878** | **0.550** | **0.708** | **0.690** | **0.571** | **0.669** |

Table 4: Accuracy by model and Bloom level, with Teaching, Diet, and Caregiving domains.

## 5.5 DISCUSSION, AREAS FOR DATASET REFINEMENT AND ADAPTIVE USAGE

The results highlight several avenues for refinement. Below-chance items can be removed, as with the five flagged Diet practices, to prevent distortion. Scenario generation should ensure contextual details (e.g., gender roles, caregiving) align with intended practices. Discrimination metrics enable tuning difficulty: high $\Delta_{model}$ questions create challenging and separable tests, while moderate spreads support baselines. Bloom-level spreads can guide item expansion across underrepresented depths. Coverage and validity checks can also automate detection of localized anomalies. Our analysis draws on psychometric traditions in educational measurement (Lord, 1980; Embretson & Reise, 2000), but full Item Response Theory (IRT)—the gold standard for modeling item difficulty and discrimination (Hambleton & Swaminathan, 1991)—was infeasible. IRT requires large model pools (typical 300+) and independent answers, whereas our setting had a small number of models with correlated outputs. Future work could revisit IRT or related latent-trait models as larger and more varied model responses become available. **Case Study in a Real-World Teaching Domain.** Evaluating LLMs with end users is difficult due to human variability, cognitive limits, and organizational constraints (e.g., cost and data governance), which prevents large-scale, controlled comparisons. Under these challenges, we luckily conducted a user study with 41 higher ed instructors, comparing a fine-tuned LLaMA-14B model against an GPT-4O-Mini. The fine-tuned model produced significantly higher user engagement and more deeper discussions of concrete classroom practices, see Appendix A.5).

## 5.6 CONCLUSION

This work introduces a domain-grounded benchmark and data generation framework for evaluating LLMs. By adapting psychometric principles—treating models as students, practices as knowledge points, and Bloom levels as cognitive depth—we design exam-inspired assessments that reveal clear model separation, domain-specific Bloom effects, and highly discriminative practices. Results demonstrate the dataset's value and the promise of psychometrically informed LLM evaluation. Although tested in three domains, the framework have potential to be generalizable to other practice-based areas where reasoning depends on domain knowledge, and can be scaled through future automation of practice extraction, scenario generation, and quality control.

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

# A APPENDIX

## A.1 DATA GENERATION PROMPTS

### A.1.1 PRACTICE → SCENARIO PROMPT

At a high level, both domains use nearly identical prompt structures: a short **system prompt** setting the assistant's role, followed by a detailed **user prompt** that combines a structured practice with a generated profile (client vs. instructor vs. caregiver). The main differences arise from the domains themselves: dietary scenarios emphasize concrete food choices, neutral wording, and subtle misalignments, while teaching scenarios emphasize classroom details, instructor background.

**Justification of scenario length and design.** Prior work shows that scenario-based MCQs commonly range from 50–200 words. For example, an analysis of medical examinations reports approximately 6,139 words for 94 items—roughly 80–100 words per scenario—in widely used professional assessments Schuwirth & van der Vleuten (2004). As researchers focusing on practice-based reasoning, we aim to mirror authentic task conditions rather than test models on artificially short (e.g., 30-word) or unrealistically long (e.g., 300-word) paragraphs that do not occur in real practice. Moreover, widely used MCQ-writing guidelines (e.g., UW School of Medicine CLIME; *National Medical Journal of India*) emphasize stems that supply enough context for reasoning while avoiding "window dressing" or irrelevant detail. These sources consistently recommend approximately 100–150 words for typical application-level scenarios and 60–80 words for simpler situations. We use violation-based scenarios because education research consistently shows that error-focused evaluation ("what went wrong?") is both more commonly used in instructional practice and more effective for improving learning than praise-based evaluation. A large meta-analysis Wisniewski et al. (2020) demonstrates that feedback is most powerful when it identifies and corrects errors to close the gap between current and desired performance. Classroom and professional practice studies similarly show that corrective feedback dominates in domains requiring procedural accuracy and safety Shute (2008); Hattie & Timperley (2007). While positive feedback can support motivation Freedberg et al. (2017), it is corrective feedback that reliably drives deeper learning and skill development.

**Diet Domain.**

**System Prompt**

```
You are a helpful assistant that creates realistic diet dilemmas
based on structured dietary guidelines practices.
```

**User Prompt Template**

```
Based on this practice:
- Goal: {goal}
- Context: {context}
- Action: {action}
- Timing: {timing}
- Person Characteristic: {person_characteristic}

Client Profile:
{JSON from DietClientProfileGenerator().generate_simplified_profile()}

IMPORTANT: Show misaligned choices only through behaviors/food items,
never by explicitly stating failure.

Requirements:
1) 50--100 words, concise
2) Use concrete foods and quantities
3) Neutral language (``chooses'' not ``fails'')
4) Include mixed choices (some good, some less ideal)
5) Include temporal cues (``often,'' ``sometimes'')
```

```
6) Exclude embedded questions

Output JSON with: "scenario"  "key question from client"
```

*Profiles were generated with the* `DietClientProfileGenerator`, *including age, sex, one health condition, a primary goal, cooking habits, one food avoidance, and two sampled traits. Scenarios were rejected if they fell outside 50–100 words, used explicit failure language, lacked temporal cues, or contained embedded questions.*

**Teaching Domain.**

**System Prompt**

```
You are an expert in teaching practices and educational analysis.
You create realistic, detailed teaching dilemmas that reflect
real classroom challenges.
```

**User Prompt Template**

```
Based on this practice:
- Learning Goal: {learning_goal}
- Context: {context}
- Timing: {timing}
- Action: {action}

Instructor Profile:
{profile_summary}

OCEAN Personality Scores: {profile['personality_profile']['ocean_scores']}

Create a realistic teaching dilemma scenario where this instructor
faces challenges implementing the practice. Requirements:
1) 50--100 words (concise, complete)
2) Show natural instructor behavior (influenced by profile,
   but without naming traits explicitly)
3) Include realistic class details
   ({class_name}, {class_size} students, {experience_description})
4) Show struggle with timing/action
5) Include student behaviors and classroom details
6) No embedded questions in the scenario text
7) Do not mention OCEAN or psychological traits explicitly

Output JSON with: "scenario" "key question from instructor"
```

*Profiles were generated with the* `InstructorProfileGenerator`, *which includes an instructor name, course details (discipline, size, format), years of experience, OCEAN scores [6] with natural-language trait descriptions, and a concise narrative summary. Scenarios were rejected if they fell outside 40–120 words, contained embedded questions, explicitly referenced personality traits, included unrealistic phrasing, or duplicated earlier scenarios.*

---

[6]In psychometrics, the big five personality trait model or five-factor model (FFM)—sometimes called by the acronym OCEAN or CANOE—is the most common scientific model for measuring and describing human personality traits. The framework groups variation in personality into five separate factors, all measured on a continuous scale: openness (O) measures creativity, curiosity, and willingness to entertain new ideas. carefulness or conscientiousness (C) measures self-control, diligence, and attention to detail. extraversion (E) measures boldness, energy, and social interactivity. amicability or agreeableness (A) measures kindness, helpfulness, and willingness to cooperate. neuroticism (N) measures depression, irritability, and moodiness. https://en.wikipedia.org/wiki/Big_Five_personality_traits

### A.1.2 ANSWER LEAKAGE KEYWORD LISTS

We applied keyword-based rejection rules to block unrealistic language (e.g., absolutist or fantastical terms), explicit violation phrases (e.g., "failed to", "ignored the guideline"), struggle-based leakage cues (e.g., "unable to", "difficulty with"), direct practice terminology, and unobservable terms (e.g., reassurance) informed by domain expert's iterative feedback. Here are some examples:

```
perfect, always, never, everyone, nobody, impossible, magic, supernatural,
fantasy, did not follow, failed to, didn't do, violated, broke the
rule, ignored the guideline, disregarded, didn't implement, too much,
excessive, overdoing, struggles to, fails to, unable to, can't seem to,
difficulty with, challenges with, problems with, issues with, trouble
with, missed opportunities, anxious about, proper, knows she should,
aware that, understands that, realizes that, impulsive decisions,
regretting their choices, feels guilty about, realizes poor choices,
takes care of, manages her, manages his, helps her, helps his, looks
after, cares for her, cares for his, shops for her, shops for his,
makes sure her, makes sure his, ensures her, ensures his, reassured,
encouragement, support, monitor, plan, address, facilite.
```

### A.1.3 HUMAN EVALUATION OF POST-VALIDATION SCENARIO

Three domain experts rated 150 paired items generated by GPT-4o-mini and Claude Sonnet 4 with the same prompt. Experts preferred Claude in 3% of cases, GPT-4o-mini in 3%, and judged outputs indistinguishable in 94% of cases, indicating no meaningful human-detectable differences. Given expense considerations, GPT-4o-mini was used. Three blinded domain experts carefully reviewed the items and documented their reasoning when preferring one model over another, including notes on why the non-preferred option was worse and if any errors occurred.

**Qualitative Analysis of Numerical Specificity** We conducted additional analyses on expert notes and the generated scenarios to better understand these differences. Qualitatively, experts noted that the primary reason for preferring Claude in some cases was the inclusion of more detailed numerical information (e.g., age-specific details and portion sizes). Importantly, none of the responses were labeled as errors. Two examples are shown below in which Claude was preferred over GPT-4o-mini.

EXAMPLE 1: GPT-4O-MINI  Paul often enjoys cooking meals on weekends, but he finds it challenging to balance his health goals with his food preferences. Last Saturday, he prepared a quinoa salad with **a generous portion** of black beans for protein, which aligns well with his goal of maintaining bone health. However, he also decided to make a classic spaghetti dish using **two cups** of gluten-free pasta and topped it with a rich, creamy Alfredo sauce, which he enjoys despite his efforts to limit saturated fat. While he remembers to use **a small amount of** olive oil when sautéing vegetables, he often adds a sprinkle of grated Parmesan cheese on top, not considering that it could contribute to his saturated fat intake. Paul loves hosting friends and often shares his meals with them, sometimes leading him to choose comfort foods that do not align with his dietary focus.

EXAMPLE 1: CLAUDE SONNET 4  Paul, an **81-year-old** retiree with celiac disease, enjoys cooking simple meals at home while managing his bone health goals. He frequently prepares grilled chicken breast and salmon fillets, which he pairs with gluten-free quinoa and steamed vegetables. However, when cooking for his weekly poker night gatherings with friends, Paul often chooses **ground beef (80/20)** for his famous gluten-free meatballs and uses coconut oil generously when pan-frying his gluten-free breaded fish. He tends to snack on **2–3 ounces** of aged cheddar cheese alternatives and rice crackers between meals, though he avoids traditional dairy. Paul wants to maintain his bone strength through his diet but finds himself uncertain about which cooking methods and ingredient substitutions work best for his dual needs of staying gluten-free while supporting his bone health.

EXAMPLE 2: GPT-4O-MINI   Carol often enjoys cooking and tends to prepare healthy meals, like a baked salmon fillet with a side of steamed broccoli and quinoa. However, during recent visits with friends, she finds herself indulging in their offerings, such as **a handful of** potato chips and **a large slice** of pepperoni pizza. With her goal to manage cholesterol, she sometimes feels conflicted when her husband suggests ordering takeout, leading her to opt for **a small** chocolate bar for dessert instead of a fruit salad. While she attempts to balance her meals, the influence of social dining and occasional cravings complicates her adherence to maintaining appropriate caloric intake for her health.

EXAMPLE 2: CLAUDE SONNET 4   Carol, a **64-year-old** retiree managing her weight and cholesterol, often prepares heart-healthy meals at home with her husband. She frequently grills **4-oz** chicken breast portions with steamed vegetables and chooses whole grain bread over white bread for their sandwiches. However, Carol tends to avoid fresh fruits entirely due to her personal preferences, missing out on important nutrients. During their weekly dinner gatherings with friends, she accommodates everyone's preferences by serving **2-cup portions** of creamy pasta dishes and rich casseroles that she knows the group enjoys. Her flexible retirement schedule allows her to cook daily, but she sometimes feels uncertain about balancing her cholesterol management goals with creating meals that satisfy both her food preferences and social dining situations.

**Quantitative Analysis of Numerical Content**   Following the reviewer's suggestion, we examined whether these numerical differences occurred systematically and whether they were consistently considered meaningful. As shown in Table 5, we used Python to extract numerical content from the scenarios and found that Claude Sonnet 4 statistically generated more numerical details overall. However, a higher quantity of numerical references did not consistently correspond to higher-quality scenarios; for example, qualitative descriptors such as "a spoon of" were often comparable to numeric expressions such as "1-oz" when interpreted in context, and age numbers in most cases does not contribute value to scenarios. Accordingly, while the relative lack of numerical detail in GPT-4o-mini made it less optimal in some edge cases, it did not render the responses incorrect or lead to substantial qualitative differences.

Table 5: Comparison of Numerical Content Across Models and Domains

| Domain | 4o Mini | Sonnet 4 | Diff | $t$ | $p$ | $d$ | $N$ |
|---|---|---|---|---|---|---|---|
| Diet | 0.96 (1.40) | 3.08 (1.62) | -2.12 | -6.99 | <0.001 | 1.40 | 50 |
| Caregiving | 0.00 (0.00) | 0.94 (0.93) | -0.94 | -7.11 | <0.001 | 1.44 | 50 |
| Teaching | 0.94 (0.47) | 2.28 (1.17) | -1.34 | -7.75 | <0.001 | 1.51 | 50 |
| All Domains | 0.63 (0.96) | 2.11 (1.54) | -1.48 | -11.42 | <0.001 | 1.15 | 150 |

A.1.4   SCENARIO → MCQ PROMPT

The initial MCQs were generated by pairing each scenario with one violated practice (the correct answer) and four/three randomly sampled distractor practices from the same domain. No additional prompting was used for distractor selection beyond randomization. To support multiple levels of cognitive difficulty, each MCQ was expanded into four versions (*Remember, Understand, Apply, Analyze*). In this step, the original scenario text was preserved, and the correct practice ID remained unchanged. The LLM was prompted to reframe the question stem and to rewrite the answer options according to Bloom's level using the practice description, rather than tailoring them directly to the scenario text. Thus, enrichment produced cognitively varied items while keeping the scenario constant.

**System Prompt.**

```
You are a helpful assistant that rewrites multiple choice questions
```

to align with Bloom's taxonomy levels. Keep the same correct practice ID and option labels. Make the language clear and educationally focused.

**User Prompt Template (Diet Domain).** For each Bloom level, a tailored instruction was used to reframe the MCQ options:

- **Remember:**

```
Rewrite each option as: "People should [specific action] to [achieve the goal]."
Use details from the practice's full_description.
```

- **Understand:**

```
Rewrite each option as a 1{2 sentence explanation of
why this practice matters for health. Use learning_goal
and impact_if_not_followed.
```

- **Apply:**

```
Rewrite each option as a 1--2 sentence description of how
this practice helps solve the health problem. Be concrete
about what to do and when.
```

- **Analyze:**

```
Rewrite each option as a 1--2 sentence analysis of why
this practice is important. Compare benefits and limitations.
```

**User Prompt Template (Teaching Domain).** The same structure was applied:

- **Remember:**

```
"Instructors should [specific action] to [achieve the goal]."
```

- **Understand:**

```
Explain why the practice matters for the problem, using
learning_goal and impact_if_not_followed.
```

- **Apply:**

```
Describe how the practice solves the teaching problem,
including when and how to implement it.
```

- **Analyze:**

```
Analyze why this practice is most important, highlighting
strengths and possible limitations.
```

**Validation.** Revised MCQs were discarded if they: Altered the correct answer label or practice ID, Produced fewer or more than five options, Introduced irrelevant or unrealistic distractors, Contained overly abstract or evaluative wording (e.g., "best practice always").

A.1.5 EVALUATION PROMPTS

We evaluated enriched MCQs using various API calls, with the same procedure across both domains (Diet and Teaching). Each MCQ was presented as a scenario, a question stem, and multiple labeled options. The model was instructed to return only the letter of its choice.

**System Prompt.** The domain-specific role was set as follows:

**Diet Domain**

```
You are an expert in diet assessment. Answer the multiple choice
question by selecting the correct option (A, B, C, or D).
Return only the letter of your choice.
```

**Teaching Domain**

```
You are an expert in educational assessment. Answer the multiple
choice question by selecting the correct option (A, B, C, D, or E).
Return only the letter of your choice.
```

**User Prompt Template.**   Identical across domains, except that Diet items contained four options (A–D) while Teaching items contained five (A–E):

```
{scenario}

Question: {question}

Options:
A. {option_A}
B. {option_B}
C. {option_C}
D. {option_D}
E. {option_E}

Select the correct answer.
```

### A.1.6   SCENARIO → DIALOGUE PROMPT

Each scenario was expanded into a multi-turn dialogue (20–30 turns) between a learner (client in Diet; instructor in Teaching) and an expert (nutritionist vs. pedagogy specialist). The dialogue generation prompt combined (i) the scenario text, (ii) a learner profile, (iii) the relevant practice description, and (iv) explicit scaffolding phases. The system role was domain-specific ("You are an expert in nutrition and dietary counseling" vs. "You are an expert in teaching practices and educational analysis"), but the user prompt followed a shared structure:

```
Generate a multi-turn conversation (20-30 turns) between a {learner_role}
and a {domain_expert} about a {domain} dilemma.

Background:
{profile attributes: client age/goals OR instructor class/experience}

Dilemma:
Scenario: {scenario}
Relevant Practice: {practice}

Structured Scaffolding:
1. Understanding the problem (3-5 turns)
2. Exploring barriers/solutions (6-10 turns)
3. Educating and planning strategies (5-7 turns)
4. Reflection and next steps (3-4 turns)

Conversation Guidelines:
- 2-4 sentences per turn
- Learner responses authentic to profile
- Expert supportive, knowledgeable, encouraging
- End with learner having a clear, realistic plan
```

**Implementation Notes.**   We used the OpenAI API for all generation steps. Parameter settings varied by task: **Scenarios and Bloom-enriched MCQs:** temperature 0.7, top-$p = 1.0$, max tokens 512, no frequency/presence penalties. **Dialogues:** higher temperature (0.9) and larger context (max tokens 1024) to encourage variety across turns. **Evaluation:** deterministic settings (temperature 0.0, max tokens 32) to force a single option letter output. In all cases, frequency and presence penalties were set to 0. Each request was retried up to three times with exponential backoff on API errors. Random seeds were not fixed, so outputs are not bit-for-bit reproducible, but balanced sampling and validation ensured consistent coverage.

## A.2 EXTENDED DATASET STATISTICS

| Predictor | $\beta$ | SE | $z$ | $p$ |
|---|---|---|---|---|
| *Teaching* | | | | |
| Intercept | 1.997 | 0.144 | 13.89 | $< .001$ |
| GPT-4O | -0.345 | 0.068 | -5.08 | $< .001$ |
| GPT-4O-Mini | -0.131 | 0.069 | -1.90 | 0.058 |
| Kimi-K2 | -0.098 | 0.069 | -1.42 | 0.156 |
| Llama-33-70B | -1.900 | 0.069 | -27.74 | $< .001$ |
| Mixtral-8×7B | -2.760 | 0.075 | -36.89 | $< .001$ |
| Qwen-25-72B | -0.024 | 0.070 | -0.35 | 0.727 |
| Qwen-3-80B | -0.299 | 0.068 | -4.38 | $< .001$ |
| Apply | -1.385 | 0.051 | -27.19 | $< .001$ |
| Remember | -0.728 | 0.051 | -14.22 | $< .001$ |
| Understand | -1.064 | 0.051 | -20.91 | $< .001$ |
| *Diet* | | | | |
| Intercept | 0.372 | 0.181 | 2.06 | 0.040 |
| GPT-4O | 0.073 | 0.066 | 1.12 | 0.263 |
| GPT-4O-Mini | 0.117 | 0.066 | 1.78 | 0.075 |
| Kimi-K2 | 0.355 | 0.067 | 5.33 | $< .001$ |
| Llama-33-70B | -0.149 | 0.065 | -2.28 | 0.022 |
| Mixtral-8×7B | -0.297 | 0.065 | -4.58 | $< .001$ |
| Qwen-25-72B | -0.077 | 0.065 | -1.18 | 0.239 |
| Qwen-3-80B | 0.215 | 0.066 | 3.24 | 0.001 |
| Apply | 0.236 | 0.046 | 5.08 | $< .001$ |
| Remember | 0.392 | 0.047 | 8.37 | $< .001$ |
| Understand | -0.062 | 0.046 | -1.36 | 0.175 |
| *Caregiving* | | | | |
| Intercept | 1.970 | 0.229 | 8.59 | $< .001$ |
| GPT-4O | 0.222 | 0.076 | 2.91 | 0.004 |
| GPT-4O-Mini | 0.373 | 0.077 | 4.82 | $< .001$ |
| Kimi-K2 | 0.095 | 0.076 | 1.25 | 0.210 |
| Llama-33-70B | 0.183 | 0.076 | 2.41 | 0.016 |
| Mixtral-8×7B | -0.227 | 0.074 | -3.06 | 0.002 |
| Qwen-25-72B | 0.183 | 0.076 | 2.41 | 0.016 |
| Qwen-3-80B | -0.051 | 0.075 | -0.68 | 0.498 |
| Apply | -0.701 | 0.056 | -12.47 | $< .001$ |
| Remember | -0.966 | 0.056 | -17.32 | $< .001$ |
| Understand | -0.603 | 0.056 | -10.69 | $< .001$ |

Table 6: GLMM fixed-effect coefficients ($\beta$) for Model and Bloom Level by domain.

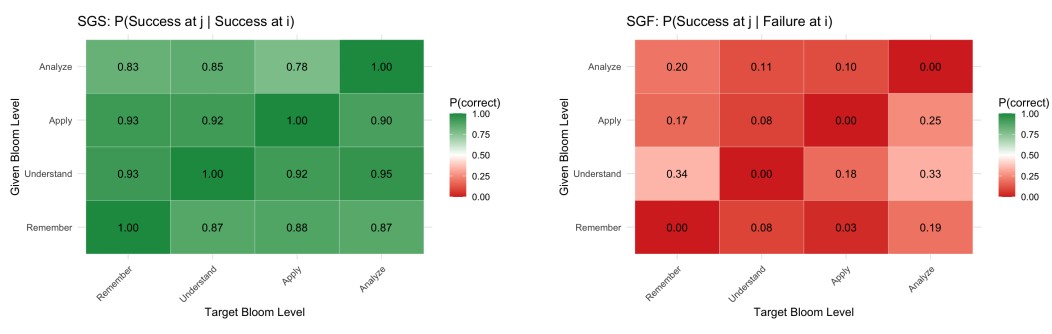

(a) SGS: Success-conditioned transitions      (b) SGF: Failure-conditioned transitions

Figure 1: Bloom Hierarchical Progression Rates by Condition.

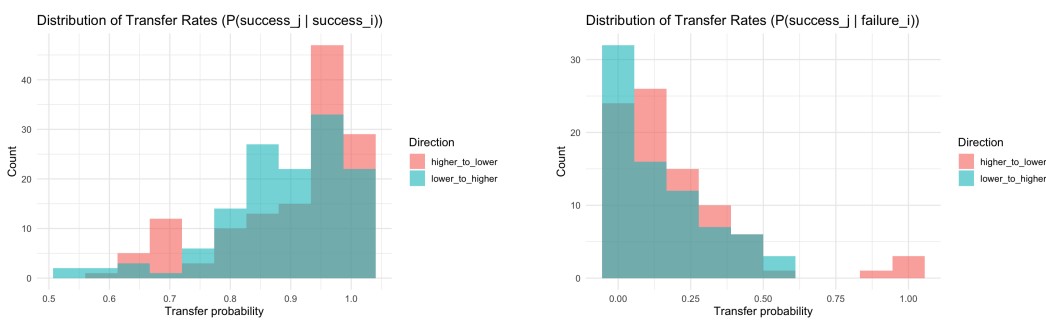

(a) SGS: Success-conditioned transitions          (b) SGF: Failure-conditioned transitions

Figure 2: Bloom Hierarchical Progression Rates by Condition and Direction.

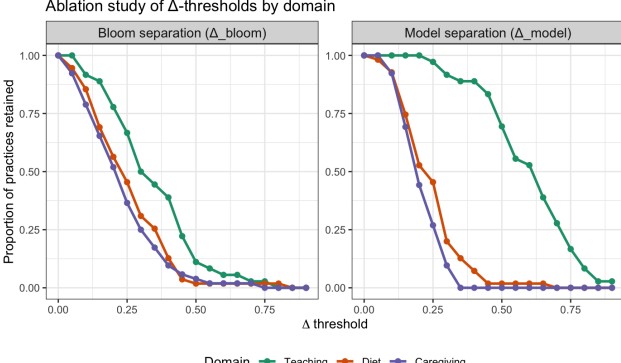

Figure 3: The figure shows how sensitive our conclusions are to the choice of $\Delta$ discrimination thresholds. Across a wide range of thresholds, the same pattern holds: Teaching practices exhibit substantially stronger discrimination than Diet or Caregiving, for both model-level separation ($\Delta_{model}$) and Bloom-level separation ($\Delta_{bloom}$). **Bloom separation ($\Delta_{bloom}$):** Teaching retains a large share of practices even at moderate $\Delta$ thresholds (0.20–0.30), whereas Diet and especially Caregiving drop sharply, indicating that Bloom-level distinctions are weakest in Caregiving. **Model separation ($\Delta_{model}$):** Teaching again retains the most practices across $\Delta$ values. Diet shows moderate discrimination, while Caregiving collapses first, indicating limited model-level spread. Overall, the ablation curves demonstrate that our findings are robust: Teaching produces higher-discrimination items, while Caregiving produces the lowest, regardless of where $\Delta$ thresholds are set.

| Domain | #Pract. | Flagged | Med. $\Delta_{model}$ | Model-sep (N,%) | Bloom-sep (N,%) | Max $\Delta$/Rank |
|---|---|---|---|---|---|---|
| Teaching | 36 | 0 | 0.714 | 35, 97.2% | 26, 72.2% | 0.000, Yes |
| Diet | 55 | 5 | 0.509 | 33, 60.0% | 20, 36.4% | 0.062, Yes |
| CareGiving | 52 | 1 | 0.433 | 30, 57.7% | 19, 36.5% | 0.019, Yes |

Table 7: Practice screening summary. "Flagged" counts best practices with baseline probability below chance. "Med. $\Delta_{model}$" is the median adjusted spread between best and worst model within a practice. "Model/Bloom-sep" counts practices showing strong model or Bloom separation. "Max $\Delta$/Rank" reports the maximum absolute change in per-model marginal means after dropping flagged items, and whether model rankings stayed identical.

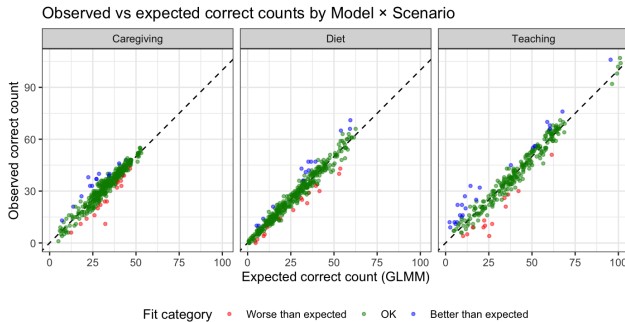

Figure 4: of observed versus expected correct counts under the GLMM for each Model×Scenario cell. Points in green indicate cells well fit by the model (OK), while blue and red highlight cells performing better or worse than expected, respectively. The dashed diagonal represents perfect fit. Separate panels show the Teaching, Diet, and Caregiving domains.

### A.3 QUALITATIVE HUMAN ANNOTATION FEEDBACK

Human review revealed that the five below-chance Diet items were nutritionally valid but became problematic due to male-centered scenario descriptions. For example, DIET11_4 ("do not exceed 1,000 µg folic acid and 45 mg iron during lactation"), DIET11_2 ("take a daily prenatal vitamin and mineral supplement"), and DIET10_9 ("seek advice from a healthcare provider regarding appropriate caloric intake during pregnancy and lactation") all target women who are pregnant or lactating. While these are legitimate recommendations, the generated scenarios often focused on male characters without clearly situating them as caregivers, or simply mentioned that they "had a family." This male-centered framing produced confusing or implausible narratives, leading to below-chance accuracy. As for Caregiving, only flagged item was CG_52 that said "*Ask your doctor or therapist if it's safe to leave your loved one alone [...] including yourself.*" It was directly relevant to the caregiver's own health; however, this perspective was never correctly generated in any scenario, with all responses focusing solely on the patient's health.

### A.4 EXAMPLE FROM TEACH-QA

**Practice ID: SM_06.**

- **Description:** Maximize the learning benefits of predictions
- **Context:** During class sessions where students are asked to make predictions about the topic of discussion
- **Timing:** Predictions made at the beginning of class should be addressed by the end of the session; those made at the close of class should be revisited at the start of the next session
- **Action:** The instructor should take note of students' predictions and ensure to address them within the specified timing, by discussing both accuracy and reasoning to reinforce learning
- **Goal:** If this practice is not followed, students may not fully understand the implications of their predictions, and the learning benefits of making predictions may not be maximized. This could lead to a lack of engagement and understanding of the subject matter",

**Generated Scenario from Practice ID: SM_06.** Dr. Lisa begins her Introduction to Political Science class by asking students to predict outcomes of current political events. With 155 students, tracking and revisiting each prediction proves overwhelming. Many students become disengaged and some forget their predictions by the time she addresses them. Despite her enthusiasm, maintaining structure and ensuring meaningful discussion becomes a challenge.

**Bloom-Enriched MCQs.**

*Remember.* Which practice is violated in this scenario?

    A. (SM_06) Maximize the learning benefits of predictions

    B. (SM_39) Extend the activity beyond class time and deepen the analysis

    C. (SM_17) Efficiently enhance learning without increasing grading workload

    D. (SM_01) Identify student misconceptions and tailor teaching accordingly

    E. (SM_50) Ensure equitable distribution of attention

**Answer:** A

*Understand.* Which practice explains why this challenge occurred?

    A. (SM_06) Without maximizing the learning benefits of predictions, students may fail to engage fully and miss critical insights

    B. (SM_39) Without extending the activity beyond class time, students may lack opportunities for deeper analysis and critical thinking

    C. (SM_17) Without integrating retrieval questions, students may struggle with retention and fail to reinforce their learning.

    D. (SM_01) Without identifying misconceptions, students may hold onto misunderstandings that hinder their learning

    E. (SM_50) Without equitable distribution of attention, some students may feel overlooked, leading to disengagement and reduced participation

**Answer:** A

*Apply.* Which practice should be used next time to address the problem?

    A. (SM_06) Address student predictions in class discussions to maximize learning benefits

    B. (SM_39) Utilize digital discussion boards to extend analysis beyond class time

    C. (SM_17) Integrate brief retrieval questions at the start or end of class to enhance learning

    D. (SM_01) Ask students for their initial thoughts on key terms to identify misconceptions

    E. (SM_50) Provide individual feedback and share insights with the entire class to ensure equitable attention

**Answer:** A

*Analyze.* Which practice best fits this scenario compared to the others?

    A. (SM_06) Helps maximize the learning benefits of predictions by engaging students in their thought processes (pro); may not work if students are unprepared to discuss their predictions or if the timing is not well managed (con).

    B. (SM_39) Helps extend learning opportunities beyond class time, fostering deeper analysis and critical thinking (pro); may not work if students lack access to digital tools or if the complexity overwhelms them (con).

    C. (SM_17) Helps enhance learning efficiency without increasing grading workload, allowing for quick feedback (pro); may not work if students do not engage with retrieval questions or if they find them too simplistic (con).

    D. (SM_01) Helps identify and address student misconceptions directly, tailoring instruction for better understanding (pro); may not work if misconceptions are deeply rooted or if students are reluctant to share their thoughts (con)

    E. (SM_50) Helps ensure equitable distribution of attention and feedback, promoting a more inclusive learning environment (pro); may not work if the feedback is too generalized and fails to address individual needs (con)

Table 8: Topics identified by LDA with average proportions of conversation content in the *FT-LLaMa* and *GPT-4O-Mini* conditions. Diff values represent the difference between conditions, with positive values indicating higher prevalence in *FT-LLaMa* and negative values indicating higher prevalence in *GPT-4O-Mini*. On average, 63.5% of the *FT-LLaMa* responses were about Topic 6 ("class, discussions, material, practice, retrieval").

| Topic | Top words | *FT-LLaMa* | *GPT-4O-Mini* | Diff | **Direction** |
|---|---|---|---|---|---|
| T6*** | class, discussions, **material**, **practice**, **retrieval** | 0.635 | 0.034 | 0.601 | FT-LLaMa ↑ |
| T1 $^{ns}$ | class, participation, engagement, share, encourage | 0.111 | 0.283 | $-0.172$ | |
| T2* | **feedback**, understanding, **concepts**, provide, **engineering** | 0.095 | 0.142 | $-0.048$ | GPT-4O-Mini ↑ |
| T3* | **data, design, concepts,** analysis, factors | 0.085 | 0.139 | $-0.054$ | GPT-4O-Mini ↑ |
| T4 $^{ns}$ | activity, class, feedback, team, online | 0.020 | 0.179 | $-0.159$ | |
| T5 $^{ns}$ | discussion, encourage, feedback, create | 0.055 | 0.223 | $-0.168$ | |

**Answer:** A

This practice was *very discrimination*, with a model separation index of $0.912$, indicating strong discrimination across models, even though the accuracy for this question was $0.727$ and not necessarily difficulty. Bloom-level performance also varied significantly, with a maximum gap of $0.163$ ($16.3$ percentage points) in accuracy between the easiest and hardest Bloom categories. This practice appeared in 17 unique scenarios, which were Bloom-enriched into 68 MCQs. With nine models answering each question, this yielded a total of 612 answers given.

## A.5 CASE STUDY IN TEACHING DOMAIN

Each instructor interacted with both systems blindly, starting from the same instructional prompt and engaging in multi-turn dialogues until their pedagogical questions were fully resolved. Below are detailed conversation log analysis using topic modeling. Three topics showed significant condition differences (Table 8). Topic 6 (*class, discussions, material, practice, retrieval*) was substantially more prevalent in *FT-LLaMa* ($W=2287$, $p_{BH}=5.19 \times 10^{-15}$, $\delta=0.950$). By contrast, Topic 2 (*feedback, understanding, concepts, provide, engineering*; $W=1577$, $p_{BH}=0.0107$, $\delta=-0.344$) and Topic 3 (*data, design, concepts, analysis, factors*; $W=1544$, $p_{BH}=0.0149$, $\delta=-0.316$) were significantly more prevalent in *GPT-4O-Mini*. The remaining topics did not differ reliably: Topic 5 ($W=1025$, $p_{BH}=0.430$, $\delta=-0.126$), Topic 1 ($W=1073$, $p_{BH}=0.567$, $\delta=-0.085$), and Topic 4 ($W=1131$, $p_{BH}=0.764$, $\delta=-0.036$); small negative deltas suggest slightly greater prevalence in *GPT-4O-Mini* but not at a significant level. We set $k=6$ as it yielded distinct, interpretable themes without over-fragmenting the data, consistent with prior topic modeling work in education and HCI. AFINN per-document normalized sentiment showed no significant tone difference between conditions (Wilcoxon $W=921$, $p=0.223$, Cliff's $\delta=-0.147$). In summary, user conversation logs in the *FT-LLaMa* version emphasized focused classroom practice (Topic 6), whereas the *GPT-4O-Mini* version distributed attention more broadly, with greater prevalence of feedback-oriented (Topic 2) and engineering/design-related (Topic 3) themes.

