# OpenReview forum: "BloomQA: Automated Benchmark Generation from Domain Guidelines Using Bloom's Taxonomy"
_ICLR.cc/2026/Conference — ICLR 2026 Conference Desk Rejected Submission_

### Official Review · Reviewer_uZA6 · 2025-10-26

**Soundness:** 2
**Presentation:** 2
**Contribution:** 2
**Rating:** 4
**Confidence:** 4

**Summary:**

BloomQA proposes a framework for generating domain-specific QA benchmarks from practice guidelines without relying on existing exam banks, demonstrating the approach in the teaching and dietetic domains to produce approximately 20,000 MCQs and 5,000 dialogs per domain. The three-stage pipeline extracts practices from guidelines using LLMs, generates violation scenarios, and expands them into Bloom's Taxonomy-aligned multiple-choice questions. Evaluation is performed with nine LLMs and shows performance separation, with top models like DeepSeek-V3 and Kimi-K2 achieving 69-75\% accuracy while weaker models approach the chance level.

However, the work suffers from fundamental weaknesses in empirical rigor and validation to be suitable for ICLR primary area "datasets and benchmarks". The framework is constructed on arbitrary choices lacking justification or ablation studies: quality thresholds (clarity $\geq 4$, similarity $\leq 2$), scenario length constraints (80-120 words), dialog turns (20-30) and discrimination thresholds ($\Delta \geq 0.2$) are presented without empirical support, and no alternative approaches are explored or compared. Most critically, only four of six Bloom levels are implemented. THe levels Evaluate and Create are omitted without explanation, thus, undermining the core claim of assessing cognitive depth and even the name of BloomQA, particularly since these highest-order skills define expert practice. Validation is severely limited: experts reviewed only 5\% of scenarios without inter-rater reliability reported, and crucially, no human baseline exists to establish whether the benchmark actually measures expertise as claimed. Psychometric evaluation relies exclusively on GLMM without comparing alternative methods such as IRT or Rasch models. The approach exhibits circular validation where LLMs generate and evaluate their own outputs, propagating systematic biases (five diet items failed due to gender bias discovered only post-hoc). Generalizability remains unsubstantiated, demonstrated only on two domains using U.S. centric guidelines, while the 60\% generation success rate and removal of nine problematic practices suggest fundamental scalability challenges.

Overall, the paper presents an interesting conceptual framework, but requires substantial empirical strengthening, human validation studies, and systematic comparison of design alternatives before its contributions can be considered robust.

**Strengths:**

1. The topic of the work is interesting and timely. As identified in the paper, it would indeed be helpful to make LLM's generate high quality MCQ in practice domains on the basis of guidelines alone.

2. The authors have attempted to use the principles of psychometric tests to evaluate their benchmark. This idea is good.

3. The use of 4 levels of Bloom's taxonomy is a strength.

**Weaknesses:**

1. The paper uses only four of six Bloom's Taxonomy levels by omitting Evaluate and Create levels without justification. These highest-order cognitive skills distinguish expert from novice practice, involving judging, critiquing, designing, and constructing novel solutions—essential for real-world teaching and dietetics practitioners. This omission fundamentally undermines the core claim of assessing ``cognitive depth" and limits the benchmark's ability to probe advanced reasoning. The MCQ format could accommodate Evaluate questions, and dialogues naturally suit Create-level tasks, making this exclusion particularly puzzling and unjustified for a framework explicitly designed around Bloom's cognitive hierarchy.

2. The paper never establishes how human subject matter experts perform on generated MCQs, creating a critical validation gap. Without human baseline data, there is no evidence that the benchmark measures domain expertise versus LLM-specific response patterns. No criterion in validity studies correlates benchmark scores with real-world competence. This absence is particularly concerning, since LLMs generated all content. More rigorous human validation would confirm whether items discriminate actual expert from novice understanding, whether difficulty levels are appropriate, and whether Bloom-level distinctions reflect genuine cognitive differences rather than mere phrasing variations.

3. The framework generates exclusively violation-based scenarios without empirical justification for the superiority or sufficiency of this approach. Real expert training also involves learning from positive examples such as effective teaching demonstrations and successful diet interventions. No evidence shows that violation-recognition correlates with practice competence or that models identifying violations can recognize correct implementations. This one-sided approach may assess only error-detection rather than full expertise scope: knowing correct actions, understanding why approaches work, and designing novel solutions. Real practitioners must choose among multiple valid approaches for contexts, not simply avoid wrong choices.

4. In addition to the above design choice, several other choices appear arbitrary (neither sufficiently justified or ablated over). Quality thresholds, scenario lengths, dialogue turns, and discrimination thresholds lack ablation studies demonstrating necessity or optimality. No comparison with alternatives are provided. Is the five W framework applicable in all domains? What are the other options? A rich discussion is missing.

5. The entire pipeline exhibits circular validation where LLMs generate practices, scenarios, questions, and answers, thereby, propagating systematic biases through every stage unchecked. The clearest evidence appears in five diet practices flagged for ``male-centered framing" in women-specific nutrition recommendations, discovered only post-hoc through statistical analysis rather than proactive bias auditing. I appreciate the authors for this discussion in the paper. An independent evaluation and identification on the number of examples to be human evaluated would add value. It is possible that the closed loop creates pattern matching to LLM generation styles rather than genuine domain knowledge assessment, as the same model class producing content also evaluates it.

6. The paper claims framework generalizability "to any domain with practice guidelines" but demonstrates only two domains (teaching and dietetics) that have characteristics that may not transfer, such as, well-defined guidelines from authoritative sources, stable best practices, and clear violation consequences. Applicability remains unproven for contested practices, rapidly-evolving fields, domains without formal guidelines, or international contexts. Moderate cross-domain correlation suggests substantial domain-specific effects, yet no analysis identifies enabling characteristics. The U.S.-centric focus (American dietary guidelines, college teaching contexts) further limits generalizability without international validation.

**Questions:**

1. How can one quantitatively estimate whether the benchmark measures domain expertise rather than LLM-specific patterns?
2. Why were evaluate and create excluded?
3. Why were violation-based scenarios only considered and not positive exemplars?
4. What steps are necessary to ensure generalization to any domain?

**Details Of Ethics Concerns:**

The dataset is US-centric. Appropriate measures to include all ethnic and linguistic groups would be ideal for such research.

---

> ### Author Response · Authors · 2025-11-23
>
> **Response to**
>
> * Missing human expert baseline on generated MCQs
>
> Practice-based domains like caregiving currently lack any established benchmark exam or human-performance dataset. We have also added a human evaluation (described below) to address this concern to our best.
>
> **Response to**
>
> * Omission of Evaluate and Create levels lacks justification
>
> We appreciate the opportunity to clarify this. As noted in our response to reviewer **cyjw**, the **Evaluate** and **Create** levels of Bloom’s Taxonomy intrinsically require **open-ended reasoning, justification, or generative production**, which cannot be validly measured using multiple-choice formats. Prior educational theory explicitly cautions that MCQs are only appropriate for the first four levels (Remember, Understand, Apply, Analyze).
>
> **Response to**
>
> * Exclusive use of violation-based scenarios questioned
>
> We use violation-based scenarios because education research consistently shows that error-focused evaluation (“what went wrong?”) is both more commonly used in instructional practice and more effective for improving learning than praise-based evaluation, we added it to Appendix A1.1.1 line 620\. Prior work shows that feedback is most powerful when it identifies and corrects errors to close the gap between current and desired performance:
>
> Wisniewski, B., Zierer, K., & Hattie, J. (2020). The power of feedback revisited: A meta-analysis of educational feedback research. *Frontiers in psychology*, *10*, 487662\.
>
> Shute, V. J. (2008). Focus on formative feedback. *Review of educational research*, *78*(1), 153-189.
>
> Hattie, J., & Timperley, H. (2007). The power of feedback. *Review of educational research*, *77*(1), 81-112.
>
> Freedberg, M., Glass, B., Filoteo, J. V., Hazeltine, E., & Maddox, W. T. (2017). Comparing the effects of positive and negative feedback in information-integration category learning. *Memory & cognition*, *45*(1), 12-25.
>
> **Response to**
>
> * Request for more human evaluation
>
> Thank you for raising this important concern. In addition, responding directly to your recommendation, During the rebuttal period, we added new human evaluation, see added human evaluation in line 265-285, line 702-710, case study with end user in Appendix 5\. Across all human evaluations, over 95% of items met expert standards.
>
> **Response to**
>
> * Lack of ablation studies for thresholds, scenario length, and dialogue structure
>
> Thank you for raising this concern. Our design choices were grounded in established guidelines for educational measurement rather than being arbitrary. We agree that scenario length and dialogue turn counts require more justification.
>
> For quality thresholds, we added clarification in line 170\. Regarding the 5W framework, the 5W structure is not domain-specific—it is a long-standing writing guideline. We use 5W to make LLM-generated scenarios complete and readable, not as a theory of domain expertise or a claim about universal applicability.
>
> For  discrimination thresholds, we added details to Step 2 in section 4 (line 325-350). The **Δ** threshold aligns with psychometric norms for identifying meaningful differences in item difficulty and discrimination. As requested by the reviewers, we **added a targeted ablation on Δ thresholds** in Appendix Figure 3 (see our detailed reply to Reviewer cyjw).
>
> For scenario lengths, dialogue turns, we noted in the methods, we used real-world recorded practitioner turns dialogues as reference points to determine naturalistic conversations length. We further added justification on wordcount in line 600-615 as it is based on long standing education research on wordcount of MCQ questions ranging from **50–200 words** in assessments. For example, an analysis of medical exams reports **\~6,139 words for \~94 items**, corresponding to roughly **80–100 words per scenario** (Lippincott).
>
> Schuwirth, L. W., & Van Der Vleuten, C. P. (2004). Different written assessment methods: what can be said about their strengths and weaknesses?. *Medical education*, *38*(9), 974-979.
>
> **Response to**
>
> * U.S.-centric focus limits international applicability
> * Question: what steps are needed to ensure true domain generalization?
>
> Thank you for the feedback, we added clarification in the paper up front Line 35 we are targeting practice based domains.  We clarify that our goal is not to generalize BloomQA to *any* possible domain, but rather to domains that share a practice-based structure—that is, domains where experts rely on clearly defined best practices, common errors, and procedural reasoning that can be represented as scenario–response pairs.
>
> In this revision, we expanded beyond the original teaching and nutrition domains by adding a new Caregiving Practices domain. See new columns and data in Table 3, 4, 6, Figure 3,4. Major Tables are directly viewable under reply to tvNd.

---

> > ### Comment · Reviewer_uZA6 · 2025-11-26
> >
> > Thank you to the authors for the responses. Reading the rebuttal and other reviews, some of the answers are not convincing (for example, violation-based criteria, non inclusion of higher levels even though convsersations are used, bias in the datset, among others). I maintain my score.

---

> > > ### Author Response · Authors · 2025-11-27
> > >
> > > Thank you for the thoughtful feedback. We would like to clarify several potential points of misunderstanding. First, our items are based on guideline-grounded error detection (“what was done wrong”), which aligns with established assessment practices in applied domains. This approach differs from domains such as mathematics, where MCQ can be easily constructed through simple right/wrong reversals.
> > >
> > > Second, we are unclear about the implied connection between conversational formats and higher-order Bloom’s levels. In our work, conversational data were used solely for fine-tuning purposes and did not involve higher-level creative tasks or constructs. The conversational data was not designed to elicit or measure higher-order cognitive skills.
> > >
> > > Third, in direct response to concerns about dataset bias (Reviewer 4f9y), we conducted additional analyses and added new results in Appendix A1.3. We appreciate the reviewer’s perspective and would welcome any further clarification that could help us strengthen the work.

---

### Official Review · Reviewer_4f9y · 2025-10-26

**Soundness:** 3
**Presentation:** 3
**Contribution:** 2
**Rating:** 4
**Confidence:** 4

**Summary:**

This paper proposes BloomQA, a framework for automatically constructing psychometrically valid evaluation benchmarks from domain knowledge guidelines, without relying on existing exam banks. Guided by Bloom’s Taxonomy, the method performs (1) LLM-assisted extraction and structuring of actionable domain practices, (2) generation of violation scenarios for quality-controlled assessment contexts, and (3) expansion into Bloom-aligned multiple-choice questions and scaffolded dialogues to support both evaluation and fine-tuning.

To validate the framework, the authors adopt a “model-as-student” paradigm, showing that BloomQA benchmarks exhibit clear cognitive-level difficulty hierarchies, effectively differentiate models with varying capabilities, and achieve strong reliability and validity. The work further releases large-scale datasets spanning multiple professional domains and demonstrates the extensibility of this approach to building practice-grounded evaluation resources.

**Strengths:**

1. The paper proposes an automated framework that extracts domain practices and converts them into implicit violation scenarios, effectively reducing dependence on existing question banks and expert-crafted items. This provides a scalable solution for constructing psychometrically meaningful benchmarks in practice-grounded domains.

2. The evaluation methodology is rooted in psychometric principles, offering reliable measurements of difficulty, discrimination, and bias. Such design ensures that the benchmark can robustly distinguish models of different capability levels while maintaining data quality and fairness.

3. The empirical analysis yields a valuable observation that LLMs exhibit cognitive-level performance patterns distinct from humans when aligned with Bloom’s hierarchy. This insight highlights the importance of domain-specific cognitive evaluation and informs future work on model analysis and improvement.

**Weaknesses:**

1. Although the benchmark creation process is grounded in real-world guideline documents, heavy reliance on LLMs introduces potential generation bias and factual inaccuracies that expert checks (limited to only 5% of scenarios) may not sufficiently capture. Moreover, the paper does not report quantitative findings from expert validation, leaving uncertainty about the actual extent and nature of residual errors.

2. The work lacks a direct comparison with existing domain-specific benchmarks, making it difficult to evaluate the practical advantages of BloomQA in terms of discrimination, coverage, or predictive relevance to real-world performance. Additional experiments contrasting model behaviors on BloomQA versus prior resources would better justify the necessity and contributions of this new benchmark.

3. While the framework is positioned as domain-agnostic and extensible, its empirical scope is restricted to psychometrically-validated MCQs in two domains. Including supplementary demonstrations in other fields (e.g., law, medicine, engineering) would provide stronger evidence for generalizability, especially given the general framing of the paper’s claims.

**Questions:**

1. The document indicates that dialogue data used for model fine-tuning can improve the performance of LLMs on higher Bloom’s taxonomy levels (e.g., Apply, Analyze). How is the generation quality of this dialogue data (such as logical coherence and domain expertise) evaluated? Has a corresponding quality scoring system been established?
2. How were the clarity and similarity thresholds in Equation 1 determined?
3. Regarding the thresholds where α = 0.05 and $\Delta_{Bloom}$ above 0.2 are often considered very meaningful, how were these thresholds determined?
4. What does β represent in Line 300?
5. The in-text equations between Line 298 and Line 318 are not numbered and need to be corrected.

---

> ### Author Response · Authors · 2025-11-23
>
> We thank you for your time and for the detailed, constructive feedback!
>
> **Response to**
>
> * More quantitative results from expert validation needed
>
> Thank you for raising this important question. We agree that validation can be further strengthened through human annotation, and in response to your suggestion we conducted an additional expert evaluation and added the results to section 3.3-Human Evaluations. Across all domains, **over 95% of sampled items met expert quality standards, and reached 95% agreement between experts.** These human-rating results have been added. Given the time constraints of the rebuttal period, this represents the maximum feasible expert evaluation, and we would be happy to extend this analysis further if needed.
>
> As described briefly in the Methods section, the data compared against three hours of real recorded expert–novice interactions to ensure structural and pedagogical realism.
>
> **Response to**
>
> * Question about how the threshold was determined
>
> Thank you for the opportunity to clarify these threshold choices. First, the value **α \= 0.05** reflects the conventional significance level widely used across statistical modeling and hypothesis testing and to avoid confusion we have revised the notation to the more familiar **p \= 0.05**.
>
> The **discrimination thresholds** follow established practice in educational measurement: as elaborated in our detailed response to Reviewer cyjw.
>
> Finally, following your suggestion, we conducted an additional ablation test to visualize the threshold decisions in Appendix Figure 3.
>
> **Response to**
>
> * How were the clarity and similarity thresholds in Equation 1 determined?
>
> The clarity and similarity thresholds in Equation 1 were designed to operationalize minimal quality standards for scenario construction based on *5W completeness* and *practice distinctiveness*.
>
> * **Clarity(bp) ≥ 4** indicates that **at least four of the five 5W elements** (who, what, when, where, why) must contain valid, non-empty content. This reflects the standard guideline—that a scenario must include sufficient contextual information to be interpretable.
>
> * **Similarity(bp) ≤ 2** ensures that **no pair of practices shares more than two identical 5W elements**, preventing near-duplicate practices and maintaining meaningful variation between items.
>
> Both thresholds were refined with domain experts, who evaluated whether different numeric cutoffs preserved meaningful distinctions between practices while maintaining readability and realism.We also acknowledge that these choices impose design constraints, and we now explicitly list them in the Limitations section.
>
> **Response to**
>
> * Concern about limited empirical scope (only two domains)
>
> Thank you for this helpful suggestion. We agree that examining performance beyond two domains can strengthen the contribution of the work. In response, we expanded our study to include a new domain on caregiving practices for caregivers, which is now fully documented in the revised manuscript. See response to reviewer tvNd
>
> **Response to**
>
> * Missing direct comparison with existing domain-specific benchmarks
>
> We would like to clarify that the domains we study are low-resource practice-based fields with no existing Bloom-aligned or scenario-based educational benchmarks. Even after incorporating the reviewer’s suggestion to broaden coverage, all three of our domains—Teaching, Dietitian Counseling, and Caregiving Practices—lack any established benchmarks or comparable datasets, which is why a one-to-one comparison with existing benchmarks is unfortunately not feasible.
>
> We fully agree that evaluating real-world relevance is important. Although such evaluation is challenging, we were able to include evidence from one domain where real-world deployment was possible. Specifically, the fine-tuned Teaching-domain model was tested in a mixed-method study with 41 higher-education instructors (added as a case study in the appendix A5). Instructors interacting with the fine-tuned version demonstrated greater engagement, more classroom-focused discussion, and more pedagogically meaningful reasoning compared with the non-fine-tuned version.
>
> **Response to**
>
> * What does β represent in Line 300?
>
> This is a standard logistic regression model for a yes/no (correct/incorrect) outcome. It estimates how the chance of a correct answer changes across models and Bloom’s levels. We’ve added the specific β values in the appendix A.2.

---

> > ### Comment · Reviewer_4f9y · 2025-11-26
> >
> > Thank you for the author's reply. I still find the explanation regarding the model-generated bias issue raised in W1 insufficient. The reported data accuracy and the results from manual verification are not fully convincing. Moreover, even a 5% error rate is non-negligible in absolute terms. (If these errors are indeed minor, could the authors provide specific comparative cases between erroneous and non-erroneous samples to demonstrate the extent of the discrepancy?)
> >
> > Furthermore, considering scalability, I believe the authors should incorporate a noise-filtering mechanism to mitigate such biases.

---

> > > ### Author Response · Authors · 2025-11-27
> > >
> > > **Response to**
> > >
> > > * More explanations on 5%
> > >
> > > We appreciate the reviewer’s thoughtful question and have expanded Appendix A1.3 to increase transparency. Three blinded domain experts carefully reviewed the items and documented their reasoning when preferring one model over another, including notes on why the non-preferred option was worse and **if any errors occurred**.
> > >
> > > We conducted additional analyses on those notes and the scenarios to better understand these differences. Qualitatively, experts noted that the primary reason for preferring Claude at times was the inclusion of more detailed numbers (e.g., age and portion sizes). None of the responses were labeled as errors. See the two examples below in which Claude was preferred over GPT-4o-mini.
> > >
> > > | GPT4o-mini | Claude Sonnet 4 |
> > > | :---- | :---- |
> > > | Paul often enjoys cooking meals on weekends, but he finds it challenging to balance his health goals with his food preferences. Last Saturday, he prepared a quinoa salad with a generous portion of black beans for protein, which aligns well with his goal of maintaining bone health. However, he also decided to make a classic spaghetti dish using **two cups** of gluten-free pasta and topped it with a rich, creamy Alfredo sauce, which he enjoys despite his efforts to limit saturated fat. While he remembers to use **a small amount** of olive oil when Sautéing vegetables, he often adds a sprinkle of grated Parmesan cheese on top, not considering that it could contribute to his saturated fat intake. Paul loves hosting friends and often shares his meals with them, sometimes leading him to choose comfort foods that don't align with his dietary focus. | Paul, an **81-year-old** retiree with celiac disease, enjoys cooking simple meals at home while managing his bone health goals. He frequently prepares grilled chicken breast and salmon fillets, which he pairs with gluten-free quinoa and steamed vegetables. However, when cooking for his weekly poker night gatherings with friends, Paul often chooses **ground beef (80/20)** for his famous gluten-free meatballs and uses coconut oil generously when pan-frying his gluten-free breaded fish. He tends to snack on **2-3 ounces** of aged cheddar cheese alternatives and rice crackers between meals, though he avoids traditional dairy. Paul wants to maintain his bone strength through his diet but finds himself uncertain about which cooking methods and ingredient substitutions work best for his dual needs of staying gluten-free while supporting his bone health. |
> > > | Carol often enjoys cooking and tends to prepare healthy meals, like a baked salmon fillet with a side of steamed broccoli and quinoa. However, during recent visits with friends, she finds herself indulging in their offerings, such as **a handful of** potato chips and **a large slice** of pepperoni pizza. With her goal to manage cholesterol, she sometimes feels conflicted when her husband suggests ordering takeout, leading her to opt for **a small** chocolate bar for dessert instead of a fruit salad. While she attempts to balance her meals, the influence of social dining and occasional cravings complicates her adherence to maintaining appropriate caloric intake for her health. | Carol, a **64-year-old** retiree managing her weight and cholesterol, often prepares heart-healthy meals at home with her husband. She frequently grills **4-oz chicken** breast portions with steamed vegetables and chooses whole grain bread over white bread for their sandwiches. However, Carol tends to avoid fresh fruits entirely due to her personal preferences, missing out on important nutrients. During their weekly dinner gatherings with friends, she accommodates everyone's preferences by serving **2-cup** portions of creamy pasta dishes and rich casseroles that she knows the group enjoys. Her flexible retirement schedule allows her to cook daily, but she sometimes feels uncertain about balancing her cholesterol management goals with creating meals that satisfy both her food preferences and social dining situations. |
> > >
> > > Following the reviewer’s suggestion, we examined whether these numerical differences occurred systematically and whether they were consistently meaningful. As shown in the table below, we extracted numerical content from the scenarios and found that Claude Sonnet 4 statistically generated more numerical details overall (see table of numbers in appendix Table 5).
> > >
> > > Meanwhile, a higher quantity of numerical details did not consistently correspond to higher-quality scenarios. For example, qualitative descriptors such as “a spoon of” were often comparable to numeric expressions such as “1 oz” when interpreted in context, and age-specific numbers were infrequently provided and i added substantive value to the scenarios. In summary, while the relative lack of numerical detail in GPT-4o-mini made it less optimal in some edge cases, it did not render the responses incorrect or lead to substantial differences.

---

### Official Review · Reviewer_tvNd · 2025-10-28

**Soundness:** 3
**Presentation:** 2
**Contribution:** 2
**Rating:** 6
**Confidence:** 2

**Summary:**

As a reviewer from the field of education, I appreciate the authors’ attempt to integrate Bloom’s taxonomy with multiple-choice questions (MCQs). However, I believe the manuscript would be stronger if it focused more clearly on educational aspects, such as applications across different disciplines, rather than broadly discussing the generation of MCQs. Therefore, I suggest removing the sections related to food and nutritionist domains from Chapter 2. The remaining parts should also be streamlined to avoid thematic inconsistency. Alternatively, the authors could focus specifically on the food and nutritionist domain instead of quickly shifting between different fields.

**Strengths:**

The integration of Bloom’s taxonomy with multiple-choice questions (MCQs) is an interesting and meaningful exploration. It holds practical value in education and could help teachers reduce their workload.

**Weaknesses:**

Although the study appears to aim for a broad exploration of QA generation (at least as suggested by the title), the target domains are confusing and the overall content lacks focus.

**Questions:**

It is recommended that the authors either concentrate their investigation on training for food and nutrition professionals or broaden their focus to include multiple disciplines within education.

**Details Of Ethics Concerns:**

The parts of the study involving human scoring or evaluation may require ethical review or approval. The authors are advised to clarify whether such procedures were reviewed by an ethics committee and to include relevant information in the manuscript if applicable.

---

> ### Author Response · Authors · 2025-11-23
>
> We greatly appreciate the time you dedicated to providing constructive and positive feedback.
>
> **Response to**
>
> * Suggest narrowing focus to food/nutrition professional training **or** broadening to multiple education domains
>
> Thank you for this helpful suggestion. We agree that examining performance beyond two domains can strengthen the contribution of the work. In this revision, we added a new **Caregiving Practices** domain, and we replicated the entire pipeline (data construction, Bloom-level design, expert validation, analysis and finetuning) across all three domains. See new columns and data in Table 3, 4, 6, Figure 3,4.
>
> For example, Table 3 can be briefly viewed here, **\*C indicates newly added Caregiving Domain (** \*T indicates old teaching Domain and  \*D indicates old diet Domain **)**
>
> | Model | R-Teach | R-Diet | R-CareG | U-Teach | U-Diet | U-CareG | A-Teach | A-Diet | A-CareG | An-Teach | An-Diet | An-CareG | Avg-T | Avg-D | Avg-C |
> | ----- | ----- | ----- | ----- | ----- | ----- | ----- | ----- | ----- | ----- | ----- | ----- | ----- | ----- | ----- | ----- |
> | DeepSeek-V3 | 0.699 | 0.646 | 0.604 | **0.730** | 0.586 | 0.723 | 0.595 | 0.610 | 0.727 | 0.904 | 0.585 | 0.825 | **0.732** | 0.607 | 0.720 |
> | GPT-4O | 0.712 | 0.657 | 0.706 | 0.608 | 0.600 | 0.734 | 0.482 | 0.620 | 0.725 | 0.879 | 0.606 | **0.840** | 0.670 | 0.621 | 0.751 |
> | GPT-4O-Mini | 0.717 | 0.700 | **0.721** | 0.662 | 0.561 | **0.771** | 0.618 | 0.653 | **0.770** | 0.850 | 0.615 | 0.828 | 0.712 | 0.632 | **0.772** |
> | Kimi-K2 | 0.692 | 0.691 | 0.675 | 0.659 | **0.612** | 0.719 | **0.620** | **0.677** | 0.704 | 0.897 | **0.695** | 0.835 | 0.717 | **0.669** | 0.733 |
> | Llama-33-70B | 0.512 | 0.629 | 0.708 | 0.299 | 0.523 | 0.735 | 0.215 | 0.595 | 0.704 | 0.427 | 0.547 | 0.833 | 0.363 | 0.574 | 0.745 |
> | Mixtral-8×7B | 0.206 | 0.648 | 0.632 | 0.206 | 0.492 | 0.700 | 0.199 | 0.571 | 0.690 | 0.213 | 0.488 | 0.710 | 0.206 | 0.550 | 0.682 |
> | Qwen-25-72B | **0.756** | 0.657 | 0.705 | 0.689 | 0.558 | 0.754 | 0.577 | 0.583 | 0.712 | **0.905** | 0.545 | 0.811 | **0.732** | 0.586 | 0.746 |
> | Qwen-3-80B-Instruct | 0.669 | **0.729** | 0.644 | 0.630 | 0.574 | 0.709 | 0.575 | 0.649 | 0.697 | 0.852 | 0.598 | 0.800 | 0.682 | 0.638 | 0.712 |
>
> For example, Table 4 can be briefly viewed here, **\*C indicates newly added Caregiving Domain**
>
> | Model | Rem-T | Rem-D | Rem-C | Und-T | Und-D | Und-C | App-T | App-D | App-C | Ana-T | Ana-D | Ana-C | Avg-T | Avg-D | Avg-C |
> | ----- | ----- | ----- | ----- | ----- | ----- | ----- | ----- | ----- | ----- | ----- | ----- | ----- | ----- | ----- | ----- |
> | Llama-3.1-8B | 0.165 | 0.317 | 0.280 | 0.065 | 0.045 | 0.318 | 0.128 | 0.183 | 0.273 | 0.290 | 0.092 | 0.303 | 0.162 | 0.159 | 0.294 |
> | Qwen2.5-7B | 0.672 | 0.638 | 0.597 | 0.690 | 0.473 | 0.637 | 0.463 | 0.563 | 0.615 | 0.845 | 0.488 | 0.690 | 0.668 | 0.541 | 0.635 |
> | Qwen2.5-14B | 0.603 | 0.638 | 0.592 | 0.590 | 0.507 | 0.602 | 0.445 | 0.565 | 0.625 | 0.840 | 0.532 | 0.692 | 0.620 | 0.561 | 0.628 |
> | FT-Llama-3.1-8B | 0.194 | 0.333 | 0.329 | 0.087 | 0.062 | 0.426 | 0.156 | 0.193 | 0.333 | 0.303 | 0.114 | 0.317 | 0.185 | 0.176 | 0.351 |
> | FT-Qwen2.5-7B | 0.662 | 0.660 | 0.598 | 0.688 | 0.497 | 0.643 | 0.487 | 0.578 | 0.627 | 0.845 | 0.513 | 0.695 | 0.671 | 0.562 | 0.641 |
> | FT-Qwen2.5-14B | 0.703 | 0.648 | 0.628 | 0.673 | 0.517 | 0.683 | 0.507 | 0.570 | 0.655 | 0.878 | 0.550 | 0.708 | 0.690 | 0.571 | 0.669 |
>
> **Response to**
>
> * Human scoring/evaluation may require ethics (IRB) review
>
> Thank you for raising this important point. We confirm that all components of the study involving human experts were conducted under approved ethical procedures. Specifically, the expert evaluation and scoring activities were reviewed and approved by our institution’s Institutional Review Board (IRB). All experts provided informed consent and were offered at a rate of approximately **USD $50 per hour** for their time and professional expertise . We have added this information to the revised manuscript’s line 118\.

---

### Official Review · Reviewer_cyjw · 2025-11-02

**Soundness:** 3
**Presentation:** 3
**Contribution:** 2
**Rating:** 4
**Confidence:** 4

**Summary:**

Bloom QA is a benchmark for open-ended question answering in applied domains, particularly focusing on education and food and diet. It has three contributions: 1) BloomQA framework that transforms domain guidelines into psychometrically-validated MCQs, 2) two datasets of Teach-QA and Diet-QA, each of which is validated in consultation with domain experts, and 3) empirical validation that rigorously evaluated the quality of the benchmark by looking into 1. alignment with Bloom’s taxonomy 2. identification of weak vs strong models (fair test in the sense of difficulty) 3. balance of the benchmark (equal percentage of different Bloom tasks/areas).

**Strengths:**

The paper took a novel initiative in turning LLM evaluation into a role-play scenario where models are students who are cognitively involved in some applied domain. The choice of education and Bloom's taxonomy inspiration has been interesting. The paper has a clear and psychometrics-informed evaluation in the later sections of the paper and looks at different combinations of scenarios such as Model × Domain, Model×Practice, and Model×Practice×Bloom.
Bloom-aligned multiple-choice options help identify different cognitive depth and engagement within the same scenario.

**Weaknesses:**

Although the paper has an innovative way of defining and evaluating the benchmark, there are still some questions that can be addressed.

- The paper should state early (Abstract/Intro) that MCQs are auto-graded, not by an LLM judge. Right now, this is only explicit in the evaluation appendix and dataset description. Also, it is unclear how the MCQ and multi-turn scenarios are evaluated. There are some mentions of rejection policies, but how does the LLM-human collaboration work together in this approach?
- Scenarios/dialogues are generated with GPT-4o-mini. The paper should discuss whether using an OpenAI family model to author data could advantage (or disadvantage) related families at evaluation time. A short bias-scan is useful, but a family-holdout (“author one family, test on others”) would strengthen the claim.
- Only four levels of Bloom's taxonomy, of all six, are used. It also does not analyze how performance at lower levels conditions performance at higher ones, or justify excluding Evaluate/Create for the benchmark. Also, evaluation across different Bloom taxonomy levels is conducted in a flat, rather than hierarchical, structure.
- The discrimination thresholds (∆model 0.20/0.50; ∆bloom 0.10/0.30) are labeled “heuristic”, but the practical significance choices need stronger justification.
- Minor: Some acronyms appear before the definition or are repeated.

**Questions:**

Most of the concerns are listed in the earlier sections, but clarification on the following points would be helpful:

- Since GPT-4o-mini authored scenarios/dialogues, did you run any “author-family vs. non-author-family” robustness checks?
- Why did the authors choose 4 out of 6 Bloom levels? How would you be able to use the hierarchical structure of the taxonomy in your evaluation and benchmark? I would like to know how one can infer mastery in a lower level of the taxonomy if the student fails at a higher level; there should be some connection between Bloom levels in the dataset.
- What empirical or pedagogical rationale supports ∆model≈0.20 and ∆bloom≈0.30 as “meaningful” cutoffs?

---

> ### Author Response · Authors · 2025-11-23
>
> **Response to**
>
> * Potential bias from GPT-4o-mini–generated data favoring OpenAI-family models
>
> Thank you for raising this point\! Table 3 shows that GPT-4o and GPT-4o-mini do not exhibit a systematic advantage over other models on the data generated by chatbot family model: across the 15 evaluated columns, the two models together top only 5 columns, a pattern not substantially different from what would be expected under random variation. Instead, Kimi-K2 emerges as the strongest model, leading in 5 columns, while other models lead in only 1–2 columns each.
>
> To further address your concern, we added a human-rating study during the rebuttal period in Appendix 1.3. We generated 50 scenarios and 50 dialogues per domain using GPT-4o-mini, and Claude Sonnet 4, and had three experts evaluate them. Experts preferred Claude over GPT-4o-mini in \~3% of items, preferred GPT-4o-mini over Claude in \~3%, and in \~94% of cases they could not tell the difference. Claude Sonnet 4 is 20–25× more expensive than GPT-4o-mini.
>
>  **Response to**
>
> * Request for empirical or pedagogical basis for ∆model ≈ 0.20 and ∆bloom ≈ 0.30
>
> The specific values of Δ do not influence our claims about the dataset’s usefulness in revealing meaningful differences; rather, they provide a more nuanced understanding of the magnitude of those differences. The discrimination thresholds (Δ\_model \= 0.20/0.50; Δ\_bloom \= 0.10/0.30) are derived from conventions in human educational measurement.
>
> Specifically, grade distribution data show that most U.S. students score between 80–100 (gradeinflation.com), so a 20-point gap is considered a meaningful difference, and a 50-point gap represents a very strong separation (e.g., 100 vs. 50). Similarly, IRT literature consistently treats 10–12 percentage-point changes (≈0.3–0.5 logits) as meaningful shifts in cognitive difficulty. More details can be found in line 324-350.
>
> We also added the Δ ablation curves in the Appendix Figure 3\.
>
> **Response to**
>
> * Use of only 4 of 6 Bloom levels questioned
>
> **1\. Why we selected 4 out of the 6 Bloom levels.**
> Education research has consistently shown that the two highest levels—Evaluate and Create—require open-ended reasoning, justification, or production of novel artifacts, which cannot be meaningfully assessed using multiple-choice formats (MCQs). Prior work explicitly cautions that MCQs are appropriate for the lower/middle cognitive levels (Remember, Understand, Apply, Analyze) but are inadequate for evaluating and creating (Krathwohl, 2002; Brookhart, 2010; Anderson & Krathwohl, 2001).
>
> Krathwohl, D. R. (2002). A revision of Bloom’s taxonomy: An overview. Theory Into Practice, 41(4), 212–218.
>
> Brookhart, S. M. (2010). How to assess higher-order thinking skills in your classroom. ASCD.
>
> Anderson, L. W., & Krathwohl, D. R. (Eds.). (2001). A taxonomy for learning, teaching, and assessing: A revision of Bloom’s taxonomy of educational objectives. Longman. or production tasks.
>
> To avoid overstating our benchmark, we revised title to describe our dataset as **“Bloom-informed”**
>
> **2\. New analysis: Hierarchical Progression Rate across Bloom levels.**
>
> To provide a finer-grained picture of how Bloom’s hierarchy is reflected in model behavior, we went beyond the aggregated conditional matrices and computed **transfer rates directly from the raw item-level data**, we computed two conditional matrices using item-level correctness:
>
> • **SGS:** P(success at level j | success at level i)
>  • **SGF:** P(success at level j | failure at level i)
>
> The **SGS** matrix shows how well models transfer from one Bloom level to another *when they already succeeded at level i*. The **SGF** matrix shows the probability of success at level j *when they failed level i*.
>
> SGS Table (P(success at j | success at i))
>
> **Rows \= i (given level succeeded)**
> **Columns \= j (target level)**
>
> | From \\ To | Remember | Understand | Apply | Analyze |
> | ----- | ----- | ----- | ----- | ----- |
> | Remember | 1.00 | 0.873 | 0.883 | 0.867 |
> | Understand | 0.931 | 1.00 | 0.922 | 0.951 |
> | Apply | 0.929 | 0.924 | 1.00 | 0.899 |
> | Analyze | 0.827 | 0.845 | 0.779 | 1.00 |
>
> SGF Table (P(success at j | failure at i))
>
> | From \\ To | Remember | Understand | Apply | Analyze |
> | ----- | ----- | ----- | ----- | ----- |
> | Remember | 0.000 | 0.084 | 0.030 | 0.191 |
> | Understand | 0.344 | 0.000 | 0.178 | 0.331 |
> | Apply | 0.171 | 0.080 | 0.000 | 0.250 |
> | Analyze | 0.199 | 0.109 | 0.103 | 0.000 |
>
> Revisions corresponding to this analysis appear in lines 310–315 and 382–390, with new visualizations added as Figures 1 and 2 in the appendix.
>
> **Response to**
>
> * Need early clarification that MCQs are auto-graded
>
> We have clarified in the Abstract that all MCQs are auto-graded and rewrote section 3.3 to clarify post-generation rejection and evaluation procedures. The full rejection rules are now provided in the Appendix 1.2 . Human experts only confirmed the final outputs after rule-based rejection.

---

### Note · Program_Chairs · 2026-01-17
**Submission Desk Rejected by Program Chairs**

The following references in this submission do not refer to real documents and/or have major errors in bibliographic information:

     Wenhao Qiu, Rui Zhang, Yixin Wang, Yujia Zhang, and Aohan Zhou. Medbloomeval: Evaluating large language models across multi-cognitive levels in medicine. In Proceedings of the 2024 Conference on Empirical Methods in Natural Language Processing (EMNLP 2024), 2024. URL https://openreview.net/forum?id=sgrJs?dbWC.
    Arthur R. Jensen and Richard Likert. Understanding item response theory. Educational Measurement Review, 2017.